# Blending *adversarial training* and *representation-conditional purification* via *aggregation* improves adversarial robustness

**Emanuele Ballarin** 📧                   *emanuele@ballarin.cc*
*AILab*
*University of Trieste*

**Alessio Ansuini**[*]📧        *alessio.ansuini@areasciencepark.it*
*Data Engineering Laboratory*
*AREA Science Park*

**Luca Bortolussi**[*]📧                 *lbortolussi@units.it*
*AILab*
*University of Trieste*

**Reviewed on OpenReview:** *https://openreview.net/forum?id=4OBXthYscW*

## Abstract

In this work, we propose a novel adversarial defence mechanism for image classification – CARSO – blending the paradigms of *adversarial training* and *adversarial purification* in a synergistic robustness-enhancing way. The method builds upon an adversarially-trained classifier, and learns to map its *internal representation* associated with a potentially perturbed input onto a distribution of tentative *clean* reconstructions. Multiple samples from such distribution are classified by the same adversarially-trained model, and a carefully chosen aggregation of its outputs finally constitutes the *robust prediction* of interest. Experimental evaluation by a well-established benchmark of strong adaptive attacks, across different image datasets, shows that CARSO is able to defend itself against adaptive *end-to-end white-box* attacks devised for stochastic defences. With a modest *clean* accuracy penalty, our method improves by a significant margin the *state-of-the-art* for CIFAR-10, CIFAR-100, and TINYIMAGENET-200 $\ell_\infty$ robust classification accuracy against AUTOATTACK.

## 1 Introduction

Vulnerability to adversarial attacks (Biggio et al., 2013; Szegedy et al., 2014) – *i.e.* the presence of inputs, usually crafted on purpose, capable of catastrophically altering the behaviour of high-dimensional models (Bortolussi & Sanguinetti, 2018) – constitutes a major hurdle towards ensuring the compliance of deep learning systems with the behaviour expected by modellers and users, and their adoption in safety-critical scenarios or tightly-regulated environments. This is particularly true for adversarially-*perturbed* inputs, where a norm-constrained perturbation – often hardly detectable by human inspection (Qin et al., 2019; Ballet et al., 2019) – is added to an otherwise legitimate input, with the intention of eliciting an anomalous response (Kurakin et al., 2018).

Given the widespread nature of the issue (Ilyas et al., 2019), and the serious concerns raised about the safety and reliability of models learnt from data in the lack of appropriate mitigations (Biggio & Roli, 2018), adversarial attacks have been extensively studied. However, obtaining generally robust machine learning (*ML*) systems remains a longstanding issue, and a major open challenge.

Research in the field has been driven by two opposing yet complementary efforts. On the one hand, the study of *failure modes* in existing models and defences, with the goal of understanding their origin and

---
[*]Joint supervision.

developing stronger attacks with varying degrees of knowledge and control over the target system (Szegedy et al., 2014; Goodfellow et al., 2015; Moosavi-Dezfooli et al., 2016; Tramèr et al., 2020). On the other hand, the construction of increasingly capable defence mechanisms. Although alternatives have been explored (Cisse et al., 2017; Tramèr et al., 2018; Carbone et al., 2020; Zhang et al., 2022), most of the latter is based on adequately leveraging *adversarial training* (Goodfellow et al., 2015; Madry et al., 2018; Tramèr & Boneh, 2019; Rebuffi et al., 2021; Gowal et al., 2021; Jia et al., 2022; Singh et al., 2023; Wang et al., 2023; Cui et al., 2023; Peng et al., 2023), *i.e.* training a *ML* model on a dataset composed of (or enriched with) adversarially-perturbed inputs associated with their correct, *pre-perturbation* labels. In fact, adversarial training has been the only technique capable of consistently providing an acceptable level of defence (Gowal et al., 2020), while still incrementally improving up to the current *state-of-the-art* (Cui et al., 2023; Peng et al., 2023; Bartoldson et al., 2024).

Another defensive approach is that of *adversarial purification* (Shi et al., 2021; Yoon et al., 2021), where a generative model is used – similarly to denoising – to recover a perturbation-free version of the input before classification is performed. Nonetheless, such attempts have generally fallen short of expectations due to inherent limitations of the generative models used in early attempts (Nie et al., 2022), or due to decreases in robust accuracy[1] when attacked *end-to-end* (Gu & Rigazio, 2015; Lee & Kim, 2024) – resulting in subpar robustness if the defensive structure is known to the adversary (Tramèr et al., 2020). More recently, the rise of diffusion-based generative models (Huang et al., 2021) and their use for purification have enabled more successful results of this kind (Nie et al., 2022; Chen et al., 2023) – although at the cost of longer training and inference times, and a much brittler robustness evaluation (Chen et al., 2023; Lee & Kim, 2024).

In this work, we design a novel adversarial defence for supervised image classification, dubbed CARSO (*Counter-Adversarial Recall of Synthetic Observations*). The approach relies on an adversarially-trained classifier (called hereinafter simply *the classifier*), endowed with a stochastic generative model (called hereinafter *the purifier*). Upon classification of a potentially-perturbed input, the latter learns to generate – from the tensor[2] of (pre)activations registered at neuron level in the former – samples from a distribution of plausible, perturbation-free reconstructions. At inference time, some of these samples are classified by the very same *classifier*, and the original input is robustly labelled by aggregating its many outputs in the form of a normalised doubly-exponential logit product. This method – to the best of our knowledge the first attempt to organically merge the *adversarial training* and *purification* paradigms – avoids the vulnerability pitfalls typical of the mere stacking of a purifier and a classifier (Gu & Rigazio, 2015; Lee & Kim, 2024), while still being able to take advantage of independent incremental improvements to adversarial training or generative modelling.

An empirical assessment[3] of the defence in the $\ell_\infty$ *white-box* setting is provided, using a *conditional* (Sohn et al., 2015; Yan et al., 2016) *variational autoencoder* (Kingma & Welling, 2014; Rezende et al., 2014) as the purifier and existing *state-of-the-art* adversarially pre-trained models as classifiers. Such choices are meant to give existing approaches – and the *adversary* attacking our architecture *end-to-end* as part of the assessment – the strongest advantage possible. Yet, in all scenarios considered, CARSO improves significantly the robustness of the pre-trained classifier – even against attacks specifically devised to fool stochastic defences like ours. Remarkably, with a modest *clean* accuracy penalty, our method improves by a significant margin the current *state-of-the-art* for CIFAR-10 (Krizhevsky, 2009), CIFAR-100 (Krizhevsky, 2009), and TINYIMAGENET-200 (Chrabaszcz et al., 2017) $\ell_\infty$ robust classification accuracy against AUTOATTACK (Croce & Hein, 2020a).

In summary, the paper makes the following contributions:

- The description of CARSO, a novel adversarial defence method synergistically blending *adversarial training* and *adversarial purification*, thanks to *representation-conditional* purification and a dedicated *robust aggregation* strategy;

---

[1] The *test set accuracy* of the frozen-weights trained classifier – computed on a dataset entirely composed of adversarially-perturbed examples generated against that specific model.

[2] Which we call *internal representation*.

[3] Implementation of the method and code for the experiments (based on *PyTorch* (Paszke et al., 2019), AdverTorch (Ding et al., 2019), and ebtorch (Ballarin, 2025)) can be found at: https://github.com/emaballarin/CARSO.

- A collection of relevant technical details fundamental to its successful training and use, originally developed for the *purifier* being a *conditional variational autoencoder* – but applicable to more general scenarios as well;

- An experimental assessment of the method, against standardised benchmark adversarial attacks – showing higher robust accuracy *w.r.t.* to existing *state-of-the-art* adversarial training and purification approaches.

The rest of the manuscript is structured as follows. In section 2 we provide an overview of selected contributions in the fields of *adversarial training* and *purification-based* defences – with focus on image classification. In section 3, an introduction is given to two integral parts of our experimental assessment: PGD adversarial training and conditional variational autoencoders. Section 4 is devoted to the intuition behind CARSO, its architectural description, and the relevant technical details that allow it to work effectively. Section 5 contains details about the experimental setup, results, comments, and limitations. Section 6 concludes the paper and outlines directions of future development.

## 2 Related work

***Adversarial training* as a defence** The idea of training a model on adversarially-generated examples as a way to make it more robust can be traced back to the very beginning of research in the area. In their seminal work, Szegedy et al. (2014) propose to perform training on a mixed collection of *clean* and adversarial data, generated beforehand.

The introduction of FGSM (Goodfellow et al., 2015) enables the efficient generation of adversarial examples along the training, with a single normalised gradient step. Its iterative generalisation PGD (Madry et al., 2018) – discussed in section 3 – significantly improves the effectiveness of the adversarial examples produced, making it still the *de facto* standard for the synthesis of adversarial training inputs (Gowal et al., 2020). Further incremental improvements have also been developed, some focused specifically on robustness assessment (*e.g.* stepsize-adaptive variants, as by Croce & Hein (2020a)).

The most recent adversarial training protocols further rely on synthetic data to increase the numerosity of training datapoints (Gowal et al., 2021; Rebuffi et al., 2021; Wang et al., 2023; Cui et al., 2023; Peng et al., 2023; Bartoldson et al., 2024), and adopt adjusted loss functions to balance robustness and accuracy (Zhang et al., 2019a) or generally foster the learning process (Cui et al., 2023). The entire model architecture may also be tuned specifically for the sake of robustness enhancement (Peng et al., 2023). At least some of such ingredients are often required to reach the current *state-of-the-art* in robust accuracy via adversarial training.

***Purification* as a defence** Amongst the first attempts of *purification-based* adversarial defence, Gu & Rigazio (2015) investigate the use of denoising autoencoders (Vincent et al., 2008) to recover examples free from adversarial perturbations. Despite its effectiveness in the denoising task, the method may indeed *increase* the vulnerability of the system when attacks are generated against it *end-to-end*. The contextually proposed improvement adds a smoothness penalty to the reconstruction loss, partially mitigating such downside (Gu & Rigazio, 2015). Similar in spirit, Liao et al. (2018) tackle the issue by computing the reconstruction loss between the last-layers representations of the frozen-weights attacked classifier, respectively receiving, as input, the *clean* and the tentatively *denoised* example.

In the work by Samangouei et al. (2018), *Generative Adversarial Networks* (GANs) (Goodfellow et al., 2014) learnt on *clean* data are used at inference time to find a plausible synthetic example – close to the perturbed input – belonging to the unperturbed data manifold. Despite encouraging results, the delicate training process of GANs and the existence of known failure modes (Zhang et al., 2018) limit the applicability of the method. More recently, a similar approach (Hill et al., 2021) employing *energy-based models* (LeCun et al., 2006) suffered from poor sample quality (Nie et al., 2022).

Purification approaches based on (conditional) variational autoencoders include the works by Hwang et al. (2019) and Shi et al. (2021). Very recently, a technique combining variational manifold learning with a test-time iterative purification procedure has also been proposed (Yang et al., 2024).

Finally, already-mentioned techniques relying on *score-* (Yoon et al., 2021) and *diffusion-* based (Nie et al., 2022; Chen et al., 2023) models have also been developed, with generally favourable results – often balanced in practice by longer training and inference times, and a much more fragile robustness assessment (Chen et al., 2023; Lee & Kim, 2024).

## 3 Preliminaries

**PGD adversarial training**  The task of finding model parameters robust to adversarial perturbations is framed by Madry et al. (2018) as a *min-max* optimisation problem seeking to minimise *adversarial risk*. The inner optimisation (*i.e.*, the generation of worst-case adversarial examples) is solved by an iterative algorithm – *Projected Gradient Descent* – interleaving gradient ascent steps in input space with the eventual projection on the shell of an $\epsilon$-ball centred around an input datapoint, thus imposing a perturbation strength constraint.

In this manuscript, we will use the shorthand notation $\epsilon_p$ to denote $\ell_p$ norm-bound perturbations of maximum magnitude $\epsilon$.

**(Conditional) variational autoencoders**  Variational autoencoders (*VAE*s) (Kingma & Welling, 2014; Rezende et al., 2014) allow the learning from data of approximate generative latent-variable models of the form $p(\boldsymbol{x}, \boldsymbol{z}) = p(\boldsymbol{x} \,|\, \boldsymbol{z})p(\boldsymbol{z})$, whose likelihood and posterior are approximately parametrised by deep artificial neural networks (*ANN*s). The problem is cast as the maximisation of a variational lower bound.

In practice, optimisation is performed iteratively – on a loss function ($\mathcal{L}_{\text{VAE}}$) given by the linear mixture of data-reconstruction loss and empirical *KL* divergence *w.r.t.* a chosen prior, computed on mini-batches of data.

*Conditional* Variational Autoencoders (Sohn et al., 2015; Yan et al., 2016) extend *VAE*s by attaching a *conditioning tensor* $\boldsymbol{c}$ – expressing specific characteristics of each example – to both $\boldsymbol{x}$ and $\boldsymbol{z}$ during training. This allows the learning of a decoder model capable of conditional data generation.

## 4 Structure of CARSO

The core ideas informing the design of our method are driven more by *first principles* rather than arising from specific contingent requirements. This section discusses such ideas, the architectural details of CARSO, and a group of technical aspects fundamental to its training and inference processes.

### 4.1 Architectural overview and principle of operation

From an architectural point of view, CARSO is essentially composed of two *ANN* models – a *classifier* and a *purifier* – operating in close synergy. The former is trained on a given classification task, whose inputs might be adversarially corrupted at inference time. The latter learns to generate samples from a distribution of potential input reconstructions, tentatively free from adversarial perturbations. Crucially, the *purifier* has only access to the internal representation of the *classifier* – and not even directly to the perturbed input – to perform its task.

During inference, for each input, the internal representation of the *classifier* is used by the *purifier* to synthesise a collection of tentatively unperturbed input reconstructions. Those are classified by the same *classifier*, and the resulting outputs are aggregated into a final *robust prediction*.

There are no specific requirements for the classifier, whose training is completely independent of the use of the model as part of CARSO. However, training it adversarially significantly improves the robust accuracy of the overall system (see Appendix D), also allowing it to benefit from established adversarial training techniques.

The purifier is also independent of specific architectural choices, provided it is capable of stochastic conditional data generation at inference time, with the internal representation of the classifier used as conditioning.

In the rest of the paper, we employ a *state-of-the-art* adversarially pre-trained WIDERESNET model as the classifier, and a purpose-built *conditional variational autoencoder* as the purifier, the latter operating decoder-only during inference. Such choice was driven by the deliberate intention to assess the adversarial

robustness of our method in its worst-case scenario against a *white-box* attacker, and with the least advantage compared to existing approaches based solely on adversarial training.

In fact, the decoder of a conditional VAE allows for exact algorithmic differentiability (Baydin et al., 2018) *w.r.t.* its conditioning set, thus averting the need for backward-pass approximation (Athalye et al., 2018a) in generating *end-to-end* adversarial attacks against the entire system, and preventing (un)intentional robustness inflation by gradient obfuscation (Athalye et al., 2018a). The same cannot be said (Chen et al., 2023) for more capable and modern purification models, such as those based *e.g.* on diffusive processes, whose proper robustness assessment is still in the process of being thoroughly understood (Lee & Kim, 2024).

A downside of such choice is represented by the reduced effectiveness of the decoder in the synthesis of complex data, due to well-known model limitations. In fact, we experimentally observe a modest increase in reconstruction cost for non-perturbed inputs, which in turn may limit the *clean* accuracy of the entire system. Nevertheless, we defend the need for a fair and transparent robustness evaluation, such as the one provided by the use of a VAE-based purifier, in the evaluation of any novel architecture-agnostic adversarial defence.

A diagram of the whole architecture is shown in Figure 1, and its detailed principles of operation are recapped below. Additionally, an ablation study investigating the need for either the *classifier* or the *purifier* being trained on adversarially-perturbed inputs is provided in Appendix D.

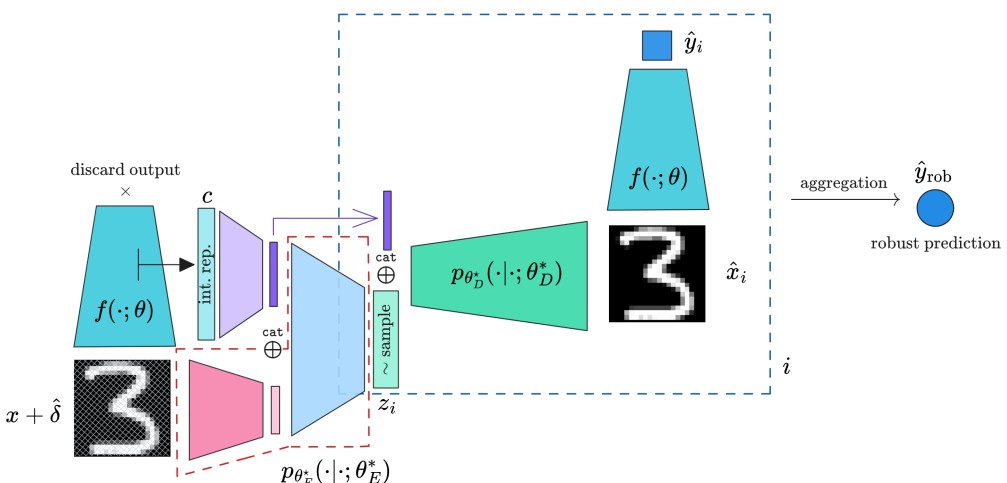

Figure 1: Schematic representation of the Carso architecture used in the experimental phase of this work. The subnetwork bordered by the red dashed line is used only during the training of the *purifier*. The subnetwork bordered by the blue dashed line is re-evaluated on different random samples $z_i$ and the resulting individual $\hat{y}_i$ are aggregated into $\hat{y}_{\text{rob}}$. The *classifier* $f(\cdot;\boldsymbol{\theta})$ is always kept frozen; the remaining network is trained on $\mathcal{L}_{\text{VAE}}(\boldsymbol{x},\hat{\boldsymbol{x}})$. More precise details on the functioning of the networks are provided in subsection 4.1.

**Training**   At training time, adversarially-perturbed examples are generated against the *classifier*, and fed to it. The tensors containing the *classifier* (pre)activations across the network are then extracted. Finally, the conditional *VAE* serving as the *purifier* is trained on perturbation-free input reconstruction, conditional on the corresponding previously-extracted internal representations, and using pre-perturbation examples as targets.

Upon completion of the training process, the encoder network is discarded, as it will not be used for inference.

**Inference**   The example requiring classification is fed to the *classifier*. Its corresponding internal representation is extracted and used to condition the generative process described by the decoder of the *VAE*. Stochastic latent variables are repeatedly sampled from the original priors, which are given by an *i.i.d.* multivariate Standard Normal distribution. Each element in the resulting set of reconstructed inputs is classified by the

same *classifier*, and the individually predicted class logits are aggregated. The result of such aggregation constitutes the robust prediction of the input class.

Remarkably, the only link between the initial potentially-perturbed input and the resulting purified reconstructions (and thus the predicted class) happens through the *internal representation* of the classifier, which serves as a *featurisation* of the original input. The whole process is exactly differentiable *end-to-end*, and the only potential hurdle to the generation of adversarial attacks against the entire system is the stochastic nature of the decoding – which is easily tackled by *Expectation over Transformation* (Athalye et al., 2018b).

### 4.2 A first-principles justification

If we consider a trained *ANN* classifier, subject to a successful adversarial attack by means of a slightly perturbed example, we observe that – both in terms of $\ell_p$ magnitude and human perception (Bartoldson et al., 2024) – a small variation on the input side of the network is amplified to a significant amount on the output side, thanks to the layerwise processing by the model. Given the deterministic nature of such processing at inference time, we speculate that the *trace* obtained by sequentially collecting the (pre)activation values within the network, along the forward pass, constitutes a richer characterisation of such an amplification process compared to the knowledge of the input alone. Indeed, as we do, it is possible to learn a direct mapping from such featurisation of the input, to a distribution of possible perturbation-free input reconstructions – in a way that takes advantage of such characterisation.

### 4.3 Hierarchical input and internal representation encoding

Training a conditional VAE requires (Sohn et al., 2015) that the conditioning set $c$ is concatenated to the input $x$ before encoding occurs, and to the sample of latent variables $z$ right before decoding. The same is also true, with the suitable adjustments, for any conditional generative approach where the target and the conditioning set must be processed jointly.

In order to ensure the usability and scalability of CARSO across the widest range of input data and classifier models, we propose to perform such processing in a hierarchical and partially disjoint fashion between the input and the conditioning set. In principle, the encoding of $x$ and $c$ can be performed by two different and independent subnetworks, until some form of joint processing must occur. This allows to retain the overall architectural structure of the purifier, while having finer-grained control over the inductive biases (Mitchell, 1980) deemed the most suitable for the respective variables.

In the experimental phase of our work, we encode the two variables independently. The input is compressed by a multilayer convolutional neural network (CNN). The internal representation – which in our case is composed of differently sized multi-channel *images* – is processed *layer by layer* by independent multilayer CNNs (responsible for encoding local information), whose flattened outputs are finally concatenated and compressed by a fully-connected layer (modelling inter-layer correlations in the representation). The resulting compressed input and conditioning set are then further concatenated and jointly encoded by a fully-connected network (FCN).

In order to use the VAE decoder at inference time, the compression machinery for the conditioning set must be preserved after training, and used to encode the internal representations extracted. The input encoder may be discarded instead.

### 4.4 Adversarially-balanced batches

Training the purifier in representation-conditional input reconstruction requires having access to adversarially-perturbed examples generated against the classifier, and to the corresponding clean data. Specifically, we use as input a mixture of *clean* and adversarially *perturbed* examples, and the clean input as the target.

Within each epoch, the *training set* of interest is shuffled (Robbins & Monro, 1951; Bottou, 1999), and only a fixed fraction of each resulting batch is adversarially perturbed. Calling $\epsilon$ the maximum $\ell_p$ perturbation norm bound for the threat model against which the *classifier* was adversarially pre-trained, the portion of perturbed examples is generated by an even split of $\text{FGSM}_{\epsilon/2}$, $\text{PGD}_{\epsilon/2}$, $\text{FGSM}_\epsilon$, and $\text{PGD}_\epsilon$ attacks.

Any smaller subset of attack types and strengths, or a detailedly unbalanced batch composition, experimentally results in a worse-performing purification model. More details justifying such choice are provided in Appendix A.

### 4.5 Robust aggregation strategy

At inference time, many different input reconstructions are classified by the *classifier*, and the respective outputs concur to the settlement of a *robust prediction*.

Calling $l_i^\alpha$ the output logit associated with class $i \in \{1, \ldots, C\}$ in the prediction by the classifier on sample $\alpha \in \{1, \ldots, N\}$, we adopt the following aggregation strategy:

$$P_i := \frac{1}{Z} \prod_{\alpha=1}^{N} e^{e^{l_i^\alpha}}$$

with $P_i$ being the aggregated probability of membership in class $i$, $Z$ a normalisation constant such that $\sum_{i=1}^{C} P_i = 1$, and $e$ Euler's number.

Such choice produces a *robust prediction* much harder to overtake in the event that an adversary selectively targets a specific input reconstruction. A heuristic analysis of this property (subsection B.1), together with an experimental justification for using such aggregation function within Carso (subsection B.2), are given in Appendix B.

## 5 Experimental assessment

Experimental evaluation of our method is carried out in terms of *robust* and *clean* image classification accuracy within three different scenarios (*a*, *b*, and *c*), determined by different classification tasks. The *white-box* threat model with a fixed $\ell_\infty$ norm bound is assumed throughout, as it generally constitutes the most demanding setup for adversarial defences.

### 5.1 Setup

**Data**  The Cifar-10 (Krizhevsky, 2009) dataset is used in *scenario (a)*, the Cifar-100 (Krizhevsky, 2009) dataset is used in *scenario (b)*, whereas the TinyImageNet-200 (Chrabaszcz et al., 2017) dataset is used in *scenario (c)*.

**Architectures**  A WideResNet-28-10 model is used as the *classifier*, adversarially pre-trained on the respective dataset – the only difference between scenarios being the size of the inputs, and the number of output logits: 10 in *scenario (a)*, 100 in *scenario (b)*, and 200 in *scenario (c)*.

The purifier is composed of a conditional VAE, processing inputs and internal representations in a partially disjoint fashion, as explained in subsection 4.3. The input is compressed by a two-layer CNN; the internal representation is instead processed layerwise by independent CNNs (three-layered in *scenarios (a)* and *(b)*, four-layered in *scenario (c)*) whose outputs are then concatenated and compressed by a fully-connected layer. A final two-layer FCN jointly encodes the compressed input and conditioning set, after the concatenation of the two. A six-layer deconvolutional network is used as the decoder.

More precise details on all architectures are given in Appendix C.

**Outer minimisation**  In *scenarios (a)* and *(b)*, the *classifier* is trained according to Cui et al. (2023); in *scenario (c)*, according to Wang et al. (2023). *Classifiers* were always acquired as pre-trained models, using publicly available weights provided by the respective authors.

The *purifier* is trained on the *VAE* loss, using *summed pixel-wise channel-wise* binary cross-entropy as the reconstruction cost. Optimisation is performed by RAdam+Lookahead (Liu et al., 2020; Zhang et al.,

2019b) with a learning rate schedule that presents a linear warm-up, a plateau phase, and a linear annealing (Smith, 2017). To promote the learning of meaningful reconstructions during the initial phases of training, the *KL divergence* term in the VAE loss is suppressed for an initial number of epochs. Afterwards, it is linearly modulated up to its actual value, along a fixed number of epochs ($\beta$ *increase*) (Higgins et al., 2017). The initial and final epochs of such modulation are reported in Table 16.

Additional scenario-specific details are provided in Appendix C.

**Inner minimisation**  $\epsilon_\infty = {}^8/_{255}$ is set as the perturbation norm bound.

Adversarial examples against the *purifier* are obtained, as explained in subsection 4.4, by FGSM$_{\epsilon/2}$, PGD$_{\epsilon/2}$, FGSM$_\epsilon$, and PGD$_\epsilon$, in a *class-untargeted* fashion on the cross-entropy loss. In the case of PGD, gradient ascent with a step size of $\alpha = 0.01$ is used.

The complete details and hyperparameters of the attacks are described in Appendix C.

**Evaluation**  In each scenario, we report the *clean* and *robust* test-set accuracy – the latter by means of AUTOATTACK (Croce & Hein, 2020a) – of the *classifier* alone, and that of the corresponding CARSO architecture.

For the *classifier* alone, the *standard* version of AUTOATTACK (*AA*) is used: *i.e.*, the worst-case accuracy on a mixture of AUTOPGD on the cross-entropy loss (Croce & Hein, 2020a) with 100 steps, AUTOPGD on the *difference of logits ratio* loss (Croce & Hein, 2020a) with 100 steps, FAB (Croce & Hein, 2020b) with 100 steps, and the *black-box* SQUARE attack (Andriushchenko et al., 2020) with 5000 queries.

In the evaluation of the CARSO architecture, the number of reconstructed samples per input is set to 8, the logits are aggregated as explained in subsection 4.5, and the output class is finally selected as the arg max of the aggregation. Due to the stochastic nature of the *purifier*, robust accuracy is assessed by a version of AUTOATTACK suitable for stochastic defences (*randAA*) – composed of AUTOPGD on the cross-entropy and *difference of logits ratio* losses, across 20 *Expectation over Transformation* (EoT) (Athalye et al., 2018b) iterations with 100 gradient ascent steps each. In the specific case of *scenario (a)*, we also assess our method by the PGD+EoT pipeline proposed by Lee & Kim (2024), as explained in subsection 5.2.

**Computational infrastructure**  All experiments were performed on an *NVIDIA DGX A100* system. Training in *scenarios (a)* and *(c)* was run on 8 *NVIDIA A100* GPUs with 40 GB of dedicated memory each; in *scenario (b)* 4 of such devices were used. Elapsed training time for the purifier in all scenarios is reported in Table 1.

Table 1: Elapsed running time for training the *purifier* in the different scenarios considered.

| Scenario | (a) | (b) | (c) |
|---|---|---|---|
| *Elapsed training time* | 159 min | 138 min | 213 min |

## 5.2   Results and discussion

An analysis of the experimental results is provided in the subsection that follows, whereas their systematic exposition is given in Table 2. Results obtained by using deliberately worse-performing pretrained *classifiers*, as well as a broader comparison with existing adversarial defences from literature, are provided in Appendix E.

***Scenario (a)***  Comparing the robust accuracy of the *classifier* model used in *scenario (a)* (Cui et al., 2023) with that resulting from the inclusion of the same model in the CARSO architecture, we observe a $+8.4\%$ increase. This is counterbalanced by a $-5.6\%$ clean accuracy decrease. The same version of CARSO further provides a $+2.42$ robustness increase *w.r.t.* the current best AT-trained model (Bartoldson et al., 2024) that employs a $\sim 4\times$ larger WIDERESNET-96-16 model.

Table 2: Clean (results in *italic*) and adversarial (results in upright) accuracy for the different models and datasets used in the respective scenarios. The following abbreviations are used: `Scen`: scenario considered; `AT/Cl`: clean accuracy for the adversarially-pretrained model used as the *classifier*, when considered alone; `C/Cl`: clean accuracy for the Carso architecture; `AT/AA`: robust accuracy (by the means of AutoAttack) for the adversarially-pretrained model used as the *classifier*, when considered alone; `C/randAA`: robust accuracy for the Carso architecture, when attacked *end-to-end* by AutoAttack for randomised defences; `Best AT/AA`: best robust accuracy result for the respective dataset (by the means of AutoAttack), obtained by adversarial training alone (any model); `Best P/AA`: best robust accuracy result for the respective dataset (by the means of AutoAttack), obtained by adversarial purification (any model). Robust accuracies in round brackets are obtained using the PGD+EoT (Lee & Kim, 2024) pipeline, developed for diffusion-based purifiers. The best clean and robust accuracies per dataset are shown in **bold**. The clean accuracies for the models referred to in the `Best` columns are shown in Table 19 (in Appendix E).

| Scen. | Dataset | AT/Cl | C/Cl | AT/AA | C/randAA (PGD+EoT) | Best AT/AA | Best P/AA (PGD+EoT) |
|-------|---------|-------|------|-------|---------------------|------------|----------------------|
| (a) | Cifar-10 | ***0.9216*** | *0.8686* | 0.6773 | **0.7613** (0.7689) | 0.7371 | 0.7812 (0.6641) |
| (b) | Cifar-100 | ***0.7385*** | *0.6806* | 0.3918 | **0.6665** | 0.4267 | 0.4609 |
| (c) | TinyImageNet-200 | ***0.6519*** | *0.5632* | 0.3130 | **0.5356** | 0.3130 | |

In addition, our method provides a remarkable $+9.72\%$ increase in robust accuracy *w.r.t.* to the best adversarial purification approach (Lin et al., 2024), a diffusion-based purifier. However, the comparison is not as straightforward. In fact, the original paper (Lin et al., 2024) reports a robust accuracy of 78.12% using AutoAttack on the gradients obtained via the adjoint method (Nie et al., 2022). As noted in Lee & Kim (2024), such evaluation (which uses the version of AutoAttack that is unsuitable for stochastic defences) leads to a large overestimation of the robustness of diffusive purifiers. As suggested in Lee & Kim (2024), the authors of Lin et al. (2024) re-evaluate the robust accuracy according to a more suitable pipeline (PGD+EoT, whose hyperparameters are shown in Table 14), obtaining a much lower robust accuracy of 66.41%. Consequently, we repeat the same evaluation for Carso and compare the worst-case robustness amongst the two. In line with typical AT methods, and unlike diffusive purification, the robustness of Carso assessed by means of *randAA* remains lower *w.r.t.* that achieved by PGD+EoT.

***Scenario (b)*** Moving to *scenario (b)*, Carso achieves a robust accuracy increase of $+27.47\%$ *w.r.t.* the *classifier* alone (Cui et al., 2023), balanced by a $-5.79\%$ decrease in clean accuracy. Our approach also improves upon the robust accuracy of the best AT-trained model (Wang et al., 2023) (WideResNet-70-16) by $+23.98\%$. In the absence of a reliable robustness evaluation by means of PGD+EoT for the best purification-based method (Lin et al., 2024), we still obtain a $+20.25\%$ increase in robust accuracy upon its (largely overestimated) AA result.

***Scenario (c)*** In *scenario (c)*, Carso improves upon the *classifier* alone (Wang et al., 2023) (which is also the best AT-based approach for TinyImageNet-200) by $+22.26\%$. A significant clean accuracy toll is imposed by the relative complexity of the dataset, *i.e.* $-8.87\%$. In this setting, we lack any additional purification-based methods.

**Assessing the impact of *gradient obfuscation*** Although the architecture of Carso is algorithmically differentiable *end-to-end* – and the integrated diagnostics of the *randAA* routines identified no pitfalls during the assessment – we additionally guard against the eventual gradient obfuscation (Athalye et al., 2018a) induced by our method by repeating the evaluation at $\epsilon_\infty = 0.95$, verifying that the resulting robust accuracy stays below random chance (Carlini et al., 2019). Results are shown in Table 3.

### 5.3 Limitations and open problems

In line with recent research aiming at the development of robust defences against multiple perturbations (Dolatabadi et al., 2022; Laidlaw et al., 2021), our method produces a decrease in *clean* accuracy *w.r.t.* the

Table 3: Robust classification accuracy against AUTOATTACK, for $\epsilon_\infty = 0.95$, as a way to assess the (lack of) impact of *gradient obfuscation* on robust accuracy evaluation.

| Scenario | (a) | (b) | (c) |
|---|---|---|---|
| $\epsilon_\infty = 0.95$ *acc.* | $<0.047$ | $<0.010$ | $\approx 0.0$ |

original model on which it is built upon – especially in *scenario (c)* as the complexity of the classification task increases. This phenomenon is partially dependent on the choice of a VAE as the generative purification model, a requirement for the fairest evaluation possible in terms of robustness.

Yet, the issue remains open: is it possible to devise a CARSO-like architecture capable of the same – if not better – robust behaviour, which is also competitively accurate on clean inputs? Potential avenues for future research may involve the development of CARSO-like architectures in which representation-conditional data generation is obtained by means of diffusion or score-based models. Alternatively, incremental developments aimed at improving the cross-talk between the purifier and the final classifier may be pursued.

Additionally, the scalability of CARSO could be strongly improved by determining whether the internal representation used in conditional data generation may be restricted to a smaller subset of layers, while still maintaining the general robustness of the method.

Finally, a thorough investigation of the *normalised doubly-exponential logit product* aggregation strategy needs to be undertaken in order to shed some light on the specific mechanisms that lead to the much improved defensive capabilities of the system.

## 6 Conclusion

In this work, we presented a novel adversarial defence mechanism tightly integrating input purification, and classification by an adversarially-trained model – in the form of representation-conditional data purification, followed by a specific logit aggregation. Our method is able to improve upon the current *state-of-the-art* in CIFAR-10, CIFAR-100, and TINYIMAGENET $\ell_\infty$ robust classification, *w.r.t.* both *adversarial training* and *purification* approaches alone.

Such results suggest a new synergistic strategy to achieve adversarial robustness in visual tasks and motivate future research on the application of the same design principles to different models and types of data.

**Broader Impact Statement**

In this work, we investigate the use of a novel technique for the improvement of adversarial robustness in image classification models. Our method does not carry the potential for additional risks or implications in sensitive areas, in comparison to the original models used. Indeed, the goal of our technique is instead to ultimately improve and enhance the robustness of classification models to make them more trustworthy, predictable, and resistant to tampering.

**Acknowledgements**

The Authors acknowledge *AREA Science Park* for the computational resources provided. This work was partially supported by funding from the European Union *Next-Generation EU* programme (*Piano Nazionale di Ripresa e Resilienza - Missione 4 Componente 2, Investimento 1.5 D. D. 1058 23/06/2022, `ECS_00000043`*) as part of the *Interconnected Nord-Est Innovation Ecosystem* (`iNEST`).

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

## A  Justification of Adversarially-balanced batches

During the incipient phases of experimentation, preliminary tests were performed with the MNIST (LeCun & Cortes, 2010) and Fashion-MNIST (Xiao et al., 2017) datasets – using a conditional *VAE* as the *purifier*, and small FCNs or *convolutional ANN*s as the *classifiers*. Adversarial examples were generated against the adversarially pre-trained *classifier*, and tentatively denoised by the *purifier* with one sample only. The resulting recovered inputs were classified by the *classifier* and the overall accuracy was recorded.

Importantly, such tests were not meant to assess the *end-to-end* adversarial robustness of the whole architecture, but only to tune the training protocol of the *purifier*.

Generating adversarial training examples by means of PGD is considered the *gold standard* (Gowal et al., 2020) and was first attempted as a natural choice to train the purifier. However, in this case, the following phenomena were observed:

- Unsatisfactory *clean* accuracy was reached upon convergence, speculatively a consequence of the *VAE* having never been trained on *clean*-to-*clean* example reconstruction;

- Persistent vulnerability to same norm-bound FGSM perturbations was noticed;

- Persistent vulnerability to smaller norm-bound FGSM and PGD perturbations was noticed.

In an attempt to mitigate such issues, the composition of adversarial examples was adjusted to specifically counteract each of the issues uncovered. The adoption of any smaller subset of attack types or strength, compared to that described in subsection 4.4, resulted in unsatisfactory mitigation.

At that point, another problem emerged: if such an adversarial training protocol was carried out in homogeneous batches, each containing the same type and strength of attack (or none at all), the resulting robust accuracy was still partially compromised due to the homogeneous ordering of attack types and strengths across batches.

Such observations lead to the final formulation of the training protocol, detailed in subsection 4.4, which mitigates to the best the issues described so far.

## B  Justification of the *robust aggregation strategy*

The rationale leading to the choice of the specific *robust aggregation strategy* described in subsection 4.5 was an attempt to answer the following question: 'How is it possible to aggregate the results of an ensemble of classifiers in a way such that it is hard to tilt the balance of the ensemble by attacking only a few of its members?'. The same reasoning can be extended to the *reciprocal* problem we are trying to solve here, where different input reconstructions obtained from the same potentially perturbed input are classified by the same model (the *classifier*).

### B.1  Heuristic analysis

Far from providing a satisfactory answer, we can analyse the behaviour of our aggregation strategy as the logit associated with a given model and class varies across its domain, under the effect of adversarial intervention. Comparison with existing (and more popular) *probability averaging* and *logit averaging* aggregation strategies should provide a heuristic justification of our choice.

We recall our aggregation strategy:

$$P_i := \frac{1}{Z} \prod_{\alpha=1}^{N} e^{e^{l_i^\alpha}}.$$

Additionally, we recall *logit averaging* aggregation

$$P_i := \frac{1}{Z} e^{\frac{1}{N}\sum_{\alpha=1}^{N} l_i^\alpha} = \frac{1}{Z} \prod_{\alpha=1}^{N} e^{\frac{1}{N} l_i^\alpha} = \frac{1}{Z} \left( \prod_{\alpha=1}^{N} e^{l_i^\alpha} \right)^{\frac{1}{N}}$$

and *probability averaging* aggregation

$$P_i := \frac{1}{Z} \sum_{\alpha=1}^{N} \frac{e^{l_i^\alpha}}{\sum_{j=1}^{C} e^{l_j^\alpha}} = \sum_{\alpha=1}^{N} e^{l_i^\alpha} \frac{1}{Q^\alpha}$$

where $Q^\alpha = \sum_{j=1}^{C} e^{l_j^\alpha}$.

Finally, since $l_i^\alpha \in \mathbb{R}, \forall l_i^\alpha$, $\lim_{x \to -\infty} e^x = 0$ and $e^0 = 1$, we can observe that $e^{l_i^\alpha} > 0$ and $e^{e^{l_i^\alpha}} > 1, \forall l_i^\alpha$.

Now, we consider a given class $i^\star$ and the *classifier* prediction on a given input reconstruction $\alpha^\star$, and study the potential effect of an adversary acting on $l_{i^\star}^{\alpha^\star}$. This adversarial intervention can be framed in two complementary scenarios: either the class $i^\star$ is correct and the adversary aims to decrease its membership probability, or the class $i^\star$ is incorrect and the adversary aims to increase its membership probability. In any case, the adversary should comply with the $\epsilon_\infty$-boundedness of its perturbation on the input.

**Logit averaging**   In the former scenario, the product of $e^{l_i^\alpha}$ terms can be arbitrarily deflated (up to zero) by lowering the $l_{i^\star}^{\alpha^\star}$ logit only. In the latter scenario, the logit can be arbitrarily inflated, and such effect is only partially suppressed by normalisation by $Z$ (a sum of $1/N$-exponentiated terms, $N \geq 1$).

**Probability averaging**   In the former scenario, although the effect of the deflation of a single logit is bounded by $e^{l_{i^\star}^{\alpha^\star}} > 0$, two attack strategies are possible: either decreasing the value of $l_{i^\star}^{\alpha^\star}$ or increasing the value of $Q^{\alpha^\star}$, giving rise to complex combined effects. In the latter scenario, the reciprocal is possible, *i.e.* either inflating $l_{i^\star}^{\alpha^\star}$ or deflating $Q^{\alpha^\star}$. Normalisation has no effect in both cases.

**Ours**   In the former scenario, the effect of logit deflation on a single product term is bounded by $e^{e^{l_{i^\star}^{\alpha^\star}}} > 1$, thus exerting only a minimal collateral effect on the product, through a decrease of $Z$. This effectively prevents *aggregation takeover* by logit deflation. Similarly to *logit averaging*, in the latter scenario, the logit can be arbitrarily inflated. However, in this case, the effect of normalisation by $Z$ is much stronger, given its increased magnitude (addends are not $1/N$-exponentiated, $N \geq 1$).

From such a comparison, our aggregation strategy is the only one that strongly prevents *adversarial takeover* by *logit deflation*, while still defending well against perturbations targeting *logit inflation*.

### B.2   Experimental analysis

To further corroborate the choice of the specific aggregation function described in subsection 4.5, we repeat the assessment of CARSO under the same conditions described in section 5, the only difference being the use of the alternative aggregation functions analysed in subsection B.1 (*i.e. logit* and *probability averaging* aggregation). Results, in terms of both *clean* and *robust* accuracy, are shown in Table 4.

In Table 5, we additionally provide the same *clean* and *robust* accuracy assessment for the naive and non algorithmically-differentiable *majority voting* aggregation strategy. In such regard, it is important to remark that the non-differentiability of majority voting results in the vast portion ($> 99\%$) of gradient samples used by AUTOATTACK being either zero or *not-a-number*. Thus, the result of robustness assessment has to be considered unreliable – and not the mark of exceptional robustness – as almost exclusively the effect of gradient obfuscation.

As we can see, the use of alternative aggregation strategies leads to minimal variations in the *clean* accuracy attained, whereas the corresponding *robust* accuracy sharply decreases – in the case of CIFAR-10 even below random chance – as the attacks become increasingly effective (or is unreliable, as it is the case for *majority*

Table 4: Clean (results in *italic*) and adversarial (results in upright) accuracy for alternative aggregation strategies used within CARSO. The following abbreviations are used: `Scen`: scenario considered; `C/Cl (L.A.)`: clean accuracy of CARSO with *logit average* aggregation; `C/randAA (L.A.)`: robust accuracy of CARSO with *logit average* aggregation, assessed by means of AUTOATTACK for stochastic defences; `C/Cl (P.A.)`: clean accuracy of CARSO with *probability average* aggregation; `C/randAA (P.A.)`: robust accuracy of CARSO with *probability average* aggregation, assessed by means of AUTOATTACK for stochastic defences; `C/Cl`: clean accuracy of CARSO with our proposed aggregation; `C/randAA`: robust accuracy of CARSO with our proposed aggregation, assessed by means of AUTOATTACK for stochastic defences. Results from the last two columns mirror those of Table 2.

| Scen. | Dataset | C/Cl (L.A.) | C/randAA (L.A.) | C/Cl (P.A.) | C/randAA (P.A.) | C/Cl | C/randAA |
|-------|---------|-------------|-----------------|-------------|-----------------|------|----------|
| (a) | CIFAR-10 | *0.8688* | 0.0086 | *0.8688* | 0.0092 | *0.8686* | **0.7613** |
| (b) | CIFAR-100 | *0.6808* | 0.0436 | *0.6807* | 0.0439 | *0.6806* | **0.6665** |

Table 5: Clean (results in *italic*) and adversarial (results in upright) accuracy resulting from the use of the *majority vote* aggregation strategy within CARSO. The following abbreviations are used: `Scen`: scenario considered; `C/Cl (M.V.)`: clean accuracy of CARSO with *majority vote* aggregation; `C/randAA (M.V.)`: robust accuracy of CARSO with *majority vote* aggregation, assessed by means of AUTOATTACK for stochastic defences. Almost the entirety of gradient samples computed by AUTOATTACK has been deemed unreliable by integrated diagnostics, and the robust accuracy results must be considered untrustworthy.

| Scen. | Dataset | C/Cl (M.V.) | C/randAA (M.V.) |
|-------|---------|-------------|-----------------|
| (a) | CIFAR-10 | *0.8691* | 0.8602 |
| (b) | CIFAR-100 | *0.6805* | 0.6698 |

*voting*). Such results strongly corroborate the use of the *normalised doubly-exponential logit product* proposed as the aggregation strategy in subsection 4.5 and prove its central role in the overall adversarial robustness and reliability of the method.

# C   Architectural details and hyperparameters

In the following section, we provide more precise details about the architectures (subsection C.1) and hyperparameters (subsection C.2) used in the experimental phase of our work.

## C.1   Architectures

In the following subsection, we describe the specific structure of the individual parts composing the *purifier* – in the three scenarios considered. As far as the *classifier* architectures are concerned, we redirect the reader to the original articles introducing those models (*i.e.*, those by Cui et al. (2023) for *scenarios (a)* and *(b)*, Wang et al. (2023) for *scenario (c)*).

During training, before being processed by the *purifier* encoder, input examples are standardised according to the statistics of the respective training dataset.

Afterwards, they are fed to the disjoint input encoder (see subsection 4.3), whose architecture is shown in Table 6. The same architecture is used in all scenarios considered.

The original input is also fed to the *classifier*. The corresponding internal representation is extracted, preserving its layered structure. In order to improve the scalability of the method, only a subset of *classifier* layers is used instead of the whole internal representation. Specifically, for each *block* of the WIDERESNET architecture, only the first layers have been considered; two *skip connections* have also been added for good measure. The exact list of those layers is reported in Table 7.

Table 6: Architecture for the *disjoint input encoder* of the *purifier*. The same architecture is used in all scenarios considered. The architecture is represented layer by layer, from input to output, in a PyTorch-like syntax. The following abbreviations are used: `Conv2D`: 2-dimensional convolutional layer; `ch_in`: number of input channels; `ch_out`: number of output channels; `ks`: kernel size; `s`: stride; `p`: padding; `b`: presence of a learnable bias term; `BatchNorm2D`: 2-dimensional batch normalisation layer; `affine`: presence of learnable affine transform coefficients; `slope`: slope for the activation function in the negative semi-domain.

| *Disjoint Input Encoder (all scenarios)* |
| --- |
| `Conv2D(ch_in=3, ch_out=6, ks=3, s=2, p=1, b=False)` |
| `BatchNorm2D(affine=True)` |
| `LeakyReLU(slope=0.2)` |
| `Conv2D(ch_in=6, ch_out=12, ks=3, s=2, p=1, b=False)` |
| `BatchNorm2D(affine=True)` |
| `LeakyReLU(slope=0.2)` |

Table 7: *Classifier* model (WIDERESNET-28-10) layer names used as (a subset of) the *internal representation* fed to the *layerwise convolutional encoder* of the *purifier*. The names reflect those used in the model implementation.

| *All scenarios* |
| --- |
| `layer.0.block.0.conv_0` |
| `layer.0.block.0.conv_1` |
| `layer.0.block.1.conv_0` |
| `layer.0.block.1.conv_1` |
| `layer.0.block.2.conv_0` |
| `layer.0.block.2.conv_1` |
| `layer.0.block.3.conv_0` |
| `layer.0.block.3.conv_1` |
| `layer.1.block.0.conv_0` |
| `layer.1.block.0.conv_1` |
| `layer.1.block.0.shortcut` |
| `layer.1.block.1.conv_0` |
| `layer.1.block.1.conv_1` |
| `layer.1.block.2.conv_0` |
| `layer.1.block.2.conv_1` |
| `layer.1.block.3.conv_0` |
| `layer.1.block.3.conv_1` |
| `layer.2.block.0.conv_0` |
| `layer.2.block.0.conv_1` |
| `layer.2.block.0.shortcut` |
| `layer.2.block.1.conv_0` |
| `layer.2.block.1.conv_1` |
| `layer.2.block.2.conv_0` |
| `layer.2.block.2.conv_1` |
| `layer.2.block.3.conv_0` |
| `layer.2.block.3.conv_1` |

Each extracted layerwise (pre)activation tensor has the shape of a multi-channel image, which is processed – independently for each layer – by a different CNN whose individual architecture is shown in Table 8 (*scenarios (a)* and *(b)*) and Table 9 (*scenario (c)*).

Table 8: Architecture for the *layerwise internal representation encoder* of the *purifier*. The architecture shown in this table is used in *scenarios (a)* and *(b)*. The architecture is represented layer by layer, from input to output, in a PyTorch-like syntax. The following abbreviations are used: `Conv2D`: 2-dimensional convolutional layer; `ch_in`: number of input channels; `ch_out`: number of output channels; `ks`: kernel size; `s`: stride; `p`: padding; `b`: presence of a learnable bias term; `BatchNorm2D`: 2-dimensional batch normalisation layer; `affine`: presence of learnable affine transform coefficients; `slope`: slope for the activation function in the negative semi-domain. The abbreviation `[ci]` indicates the number of input channels for the *(pre)activation tensor* of each extracted layer. The abbreviation `ceil` indicates the *ceiling* integer rounding function.

| *Layerwise Internal Representation Encoder (scenarios (a) and (b))* |
|---|
| `Conv2D(ch_in=[ci], ch_out=ceil([ci]/2), ks=3, s=1, p=0, b=False)` |
| `BatchNorm2D(affine=True)` |
| `LeakyReLU(slope=0.2)` |
| `Conv2D(ch_in=ceil([ci]/2), ch_out=ceil([ci]/4), ks=3, s=1, p=0, b=False)` |
| `BatchNorm2D(affine=True)` |
| `LeakyReLU(slope=0.2)` |
| `Conv2D(ch_in=ceil([ci]/4), ch_out=ceil([ci]/8), ks=3, s=1, p=0, b=False)` |
| `BatchNorm2D(affine=True)` |
| `LeakyReLU(slope=0.2)` |

Table 9: Architecture for the *layerwise internal representation encoder* of the *purifier*. The architecture shown in this table is used in *scenario (c)*. The architecture is represented layer by layer, from input to output, in a PyTorch-like syntax. The following abbreviations are used: `Conv2D`: 2-dimensional convolutional layer; `ch_in`: number of input channels; `ch_out`: number of output channels; `ks`: kernel size; `s`: stride; `p`: padding; `b`: presence of a learnable bias term; `BatchNorm2D`: 2-dimensional batch normalisation layer; `affine`: presence of learnable affine transform coefficients; `slope`: slope for the activation function in the negative semi-domain. The abbreviation `[ci]` indicates the number of input channels for the *(pre)activation tensor* of each extracted layer. The abbreviation `ceil` indicates the *ceiling* integer rounding function.

| *Layerwise Internal Representation Encoder (scenario (c))* |
|---|
| `Conv2D(ch_in=[ci], ch_out=ceil([ci]/2), ks=3, s=1, p=0, b=False)` |
| `BatchNorm2D(affine=True)` |
| `LeakyReLU(slope=0.2)` |
| `Conv2D(ch_in=ceil([ci]/2), ch_out=ceil([ci]/4), ks=3, s=1, p=0, b=False)` |
| `BatchNorm2D(affine=True)` |
| `LeakyReLU(slope=0.2)` |
| `Conv2D(ch_in=ceil([ci]/4), ch_out=ceil([ci]/8), ks=3, s=1, p=0, b=False)` |
| `BatchNorm2D(affine=True)` |
| `LeakyReLU(slope=0.2)` |
| `Conv2D(ch_in=ceil([ci]/8), ch_out=ceil([ci]/16), ks=3, s=1, p=0, b=False)` |
| `BatchNorm2D(affine=True)` |
| `LeakyReLU(slope=0.2)` |

The resulting tensors (still having the shape of multi-channel images) are then jointly processed by a fully-connected subnetwork whose architecture is shown in Table 10. The value of `fcrepr` for the different scenarios considered is shown in Table 15.

The *compressed input* and *compressed internal representation* so obtained are finally jointly encoded by an additional fully-connected subnetwork whose architecture is shown in Table 11. The output is a tuple of means and standard deviations to be used to sample the stochastic latent code $z$.

Table 10: Architecture for the *fully-connected representation encoder* of the *purifier*. The architecture shown in this table is used in all scenarios considered. The architecture is represented layer by layer, from input to output, in a PyTorch-like syntax. The following abbreviations are used: `Concatenate`: layer concatenating its input features; `flatten_features`: whether the input features are to be flattened before concatenation; `feats_in`, `feats_out`: number of input and output features of a linear layer; `b`: presence of a learnable bias term; `BatchNorm1D`: 1-dimensional batch normalisation layer; `affine`: presence of learnable affine transform coefficients; `slope`: slope for the activation function in the negative semi-domain. The abbreviation `[computed]` indicates that the number of features is computed according to the shape of the concatenated input tensors. The value of `fcrepr` for the different scenarios considered is shown in Table 15.

---
*Fully-Connected Representation Encoder (all scenarios)*

---
```
Concatenate(flatten_features=True)
Linear(feats_in=[computed], feats_out=fcrepr, b=False)
BatchNorm1D(affine=True)
LeakyReLU(slope=0.2)
```
---

Table 11: Architecture for the *fully-connected joint encoder* of the *purifier*. The architecture shown in this table is used in all scenarios considered. The architecture is represented layer by layer, from input to output, in a PyTorch-like syntax. The following abbreviations are used: `Concatenate`: layer concatenating its input features; `flatten_features`: whether the input features are to be flattened before concatenation; `feats_in`, `feats_out`: number of input and output features of a linear layer; `b`: presence of a learnable bias term; `BatchNorm1D`: 1-dimensional batch normalisation layer; `affine`: presence of learnable affine transform coefficients; `slope`: slope for the activation function in the negative semi-domain. The abbreviation `[computed]` indicates that the number of features is computed according to the shape of the concatenated input tensors. The value of `fjoint` for the different scenarios considered is shown in Table 15. The last *layer* of the network returns a tuple of 2 tensors, each independently processed – from the output of the previous layer – by the two comma-separated *sub-layers*.

---
*Fully-Connected Joint Encoder (all scenarios)*

---
```
Concatenate(flatten_features=True)
Linear(feats_in=[computed], feats_out=fjoint, b=False)
BatchNorm1D(affine=True)
LeakyReLU(slope=0.2)
( Linear(feats_in=fjoint, feats_out=fjoint, b=True),
        Linear(feats_in=fjoint, feats_out=fjoint, b=True) )
```
---

The sampler used for the generation of such latent variables $z$, during the training of the *purifier*, is a reparameterised (Kingma & Welling, 2014) Normal sampler $z \sim \mathcal{N}(\mu, \sigma)$. During inference, $z$ is sampled by reparameterisation from the *i.i.d* Standard Normal distribution $z \sim \mathcal{N}(0, 1)$ (*i.e.* from its original prior).

The architectures for the decoder of the *purifier* are shown in Table 12 (*scenarios (a) and (b)*) and Table 13 (*scenario (c)*).

Table 12: Architecture for the decoder of the *purifier*. The architecture shown in this table is used in *scenarios (a)* and *(b)*. The architecture is represented layer by layer, from input to output, in a PyTorch-like syntax. The following abbreviations are used: `Concatenate`: layer concatenating its input features; `flatten_features`: whether the input features are to be flattened before concatenation; `feats_in`, `feats_out`: number of input and output features of a linear layer; `b`: presence of a learnable bias term; `ConvTranspose2D`: 2-dimensional transposed convolutional layer; `ch_in`: number of input channels; `ch_out`: number of output channels; `ks`: kernel size; `s`: stride; `p`: padding; `op`: PyTorch parameter '`output padding`', used to disambiguate the number of spatial dimensions of the resulting output; `b`: presence of a learnable bias term; `BatchNorm2D`: 2-dimensional batch normalisation layer; `affine`: presence of learnable affine transform coefficients; `slope`: slope for the activation function in the negative semi-domain. The values of `fjoint` and `fcrepr` for the different scenarios considered are shown in Table 15.

| Decoder (scenarios (a) and (b)) |
|---|
| ```
Concatenate(flatten_features=True)
Linear(feats_in=[fjoint+fcrepr], feats_out=2304, b=True)
LeakyReLU(slope=0.2)
Unflatten(256, 3, 3)
ConvTranspose2D(ch_in=256, ch_out=256, ks=3, s=2, p=1, op=0, b=False)
BatchNorm2D(affine=True)
LeakyReLU(slope=0.2)
ConvTranspose2D(ch_in=256, ch_out=128, ks=3, s=2, p=1, op=0, b=False)
BatchNorm2D(affine=True)
LeakyReLU(slope=0.2)
ConvTranspose2D(ch_in=128, ch_out=64, ks=3, s=2, p=1, op=0, b=False)
BatchNorm2D(affine=True)
LeakyReLU(slope=0.2)
ConvTranspose2D(ch_in=64, ch_out=32, ks=3, s=2, p=1, op=0, b=False)
BatchNorm2D(affine=True)
LeakyReLU(slope=0.2)
ConvTranspose2D(ch_in=32, ch_out=3, ks=2, s=1, p=1, op=0, b=True)
Sigmoid()
``` |

## C.2 Hyperparameters

In the following section, we provide the hyperparameters used for *adversarial example generation* and *optimisation* during the training of the *purifier*, and those related to the *purifier* model architectures. We also provide the hyperparameters for the PGD+EOT attack, which is used as a complementary tool for the evaluation of adversarial robustness.

**Attacks** The hyperparameters used for the adversarial attacks described in subsection 4.4 are shown in Table 14. The value of $\epsilon_\infty$ is fixed to $\epsilon_\infty = {}^8/_{255}$. With the only exception of $\epsilon_\infty$, AUTOATTACK is to be considered a *hyperparameter-free* adversarial example generator.

**Architectures** Table 15 contains the hyperparameters that define the model architectures used as part of the *purifier*, in the different scenarios considered.

**Training** Table 16 collects the hyperparameters governing the training of the *purifier* in the different scenarios considered.

Table 13: Architecture for the decoder of the *purifier*. The architecture shown in this table is used in *scenario (c)*. The architecture is represented layer by layer, from input to output, in a PyTorch-like syntax. The following abbreviations are used: `Concatenate`: layer concatenating its input features; `flatten_features`: whether the input features are to be flattened before concatenation; `feats_in`, `feats_out`: number of input and output features of a linear layer; `b`: presence of a learnable bias term; `ConvTranspose2D`: 2-dimensional transposed convolutional layer; `ch_in`: number of input channels; `ch_out`: number of output channels; `ks`: kernel size; `s`: stride; `p`: padding; `op`: PyTorch parameter '`output padding`', used to disambiguate the number of spatial dimensions of the resulting output; `b`: presence of a learnable bias term; `BatchNorm2D`: 2-dimensional batch normalisation layer; `affine`: presence of learnable affine transform coefficients; `slope`: slope for the activation function in the negative semi-domain. The values of `fjoint` and `fcrepr` for the different scenarios considered are shown in Table 15.

| *Decoder (scenario (c))* |
| --- |
| `Concatenate(flatten_features=True)` |
| `Linear(feats_in=[fjoint+fcrepr], feats_out=4096, b=True)` |
| `LeakyReLU(slope=0.2)` |
| `Unflatten(256, 4, 4)` |
| `ConvTranspose2D(ch_in=256, ch_out=256, ks=3, s=2, p=1, op=1, b=False)` |
| `BatchNorm2D(affine=True)` |
| `LeakyReLU(slope=0.2)` |
| `ConvTranspose2D(ch_in=256, ch_out=128, ks=3, s=2, p=1, op=1, b=False)` |
| `BatchNorm2D(affine=True)` |
| `LeakyReLU(slope=0.2)` |
| `ConvTranspose2D(ch_in=128, ch_out=64, ks=3, s=2, p=1, op=1, b=False)` |
| `BatchNorm2D(affine=True)` |
| `LeakyReLU(slope=0.2)` |
| `ConvTranspose2D(ch_in=64, ch_out=32, ks=3, s=2, p=1, op=1, b=False)` |
| `BatchNorm2D(affine=True)` |
| `LeakyReLU(slope=0.2)` |
| `ConvTranspose2D(ch_in=32, ch_out=3, ks=3, s=1, p=1, op=0, b=True)` |
| `Sigmoid()` |

Table 14: Hyperparameters for the attacks used for training and testing the *purifier* The FGSM and PDG attacks refer to the training phase (see subsection 4.4), whereas the PGD+EoT attack (Lee & Kim, 2024) refers to the robustness assessment pipeline. The entry `CCE` denotes the *Categorical CrossEntropy* loss function. The $\ell_\infty$ threat model is assumed, and all inputs are linearly rescaled within $[0.0, 1.0]$ before the attack.

|  | FGSM | PGD | PGD+EoT |
| --- | --- | --- | --- |
| Input clipping | $[0.0, 1.0]$ | $[0.0, 1.0]$ | $[0.0, 1.0]$ |
| # of steps | 1 | 40 | 200 |
| Step size | $\epsilon_\infty$ | 0.01 | 0.007 |
| Loss function | *CCE* | *CCE* | *CCE* |
| # of EoT iterations | 1 | 1 | 20 |
| Optimiser |  | *SGD* | *SGD* |

Table 15: Scenario-specific architectural hyperparameters for the *purifier*, as referred to in Table 10, Table 11, Table 12, and Table 13.

|  | *Scenario (a)* | *Scenario (b)* | *Scenario (c)* |
| --- | --- | --- | --- |
| `fcrepr` | 512 | 512 | 768 |
| `fjoint` | 128 | 128 | 192 |

Table 16: Hyperparameters used for training the *purifier*, grouped by scenario. The entry `CCE` denotes the *Categorical CrossEntropy* loss function. The *LR* scheduler is stepped after each epoch.

| | All *scenarios* | *Sc. (a)* | *Sc. (b)* | *Sc. (c)* |
|---|---|---|---|---|
| Optimiser | RADAM+LOOKAHEAD | | | |
| RADAM $\beta_1$ | 0.9 | | | |
| RADAM $\beta_2$ | 0.999 | | | |
| RADAM $\epsilon$ | $10^{-8}$ | | | |
| RADAM *Weight Decay* | None | | | |
| LOOKAHEAD *averaging decay* | 0.8 | | | |
| LOOKAHEAD steps | 6 | | | |
| Initial *LR* | $5 \times 10^{-9}$ | | | |
| Loss function | *CCE* | | | |
| Sampled reconstructions per input | 8 | | | |
| Epochs | | 200 | 200 | 250 |
| *LR* warm-up epochs | | 25 | 25 | 31 |
| *LR* plateau epochs | | 25 | 25 | 31 |
| *LR* annealing epochs | | 150 | 150 | 188 |
| Plateau *LR* | | 0.064 | 0.064 | 0.0128 |
| Final *LR* | | $4.346 \times 10^{-4}$ | $4.346 \times 10^{-4}$ | $1.378 \times 10^{-4}$ |
| $\beta$ *increase* initial epoch | | 25 | 25 | 32 |
| $\beta$ *increase* final epoch | | 34 | 34 | 43 |
| Batch size | | 5120 | 2560 | 1024 |
| Adversarial *batch fraction* | | 0.5 | 0.15 | 0.01 |

## D  Ablation study on the need for adversarial training

In order to determine whether it is necessary to train on adversarial examples each of the constituent parts of CARSO, an ablation study is performed. The architecture of CARSO provided in section 5.1 is compared in terms of *clean* and *robust* accuracy with those ablated as follows. A WIDERESNET-28-10 model is always used as the *classifier*.

- Both the initial instance of the *classifier* (that used to extract the *internal representation*) and the final (that used to actually perform classification) are trained on clean examples only.

- The final *classifier* is trained on clean examples only, whereas the former is adversarially-trained.

- The initial *classifier* is trained on clean examples only, whereas the latter is adversarially-trained.

Results of such comparison are shown in Table 17.

As it is possible to see, only the *clean/AT* ablation provides *clean* and *adversarial* accuracies comparable to that of the original CARSO architecture – and indeed, it even determines an improvement on the CIFAR-10 and CIFAR-100 datasets. On the other hand, adversarial training of the former instance of the classifier is necessary to achieve the best robustness results on TINYIMAGENET-200.

Keeping in mind that the TINYIMAGENET-200 dataset is the closest representative considered for larger-scale datasets, and that empirical *robust accuracy* results only constitute an upper bound of maximally attainable robust accuracy, we support the original CARSO architecture as the most effective for the achievement of adversarial robustness on generic image classification datasets – even if at the cost of a slight *clean* accuracy penalty. Since an adversarially-trained classifier would be used nonetheless as the latter, this does not incur in increased training-time computational intensity.

Table 17: Results of the ablation study on the architecture of Carso. The *Clean Acc.* column shows the test-set accuracy on uncorrupted inputs for the specific ablated model; the *randAA Acc.* column shows the accuracy of the same model on test-set inputs perturbed by means of the version of AutoAttack suitable for stochastic defences. In the *Type of ablation* column, any entry different from `None` (original architecture) indicates the type of training used for the first (before the solidus) and the second (after the solidus) classifier following input-to-output flow, within the ablated Carso architecture.

| Dataset | Type of ablation | Clean Acc. | randAA Acc. |
|---|---|---|---|
| Cifar-10 | *clean/clean* | *0.7314* | 0.7070 |
| | *AT/clean* | *0.6743* | < 0.7070 |
| | *clean/AT* | ***0.8892*** | **0.7975** |
| | None | *0.8686* | 0.7613 |
| Cifar-100 | *clean/clean* | *0.4395* | 0.4032 |
| | *AT/clean* | *0.4373* | < 0.4032 |
| | *clean/AT* | ***0.6876*** | **0.6716** |
| | None | *0.6806* | 0.6665 |
| TinyImageNet-200 | *clean/AT* | ***0.5677*** | 0.5281 |
| | None | *0.5632* | **0.5356** |

# E    Further results and comparisons

The following section contains additional results and comparisons, in the form of tabular data, that may be of interest to the reader.

In particular, Table 19 compares the *clean* and *robust* accuracy of Carso with those of the top-5 adversarial training defences according to the RobustBench Croce et al. (2021) leaderboard[4] and the top-5 (if available) purification-based defences according to Lee & Kim (2024), for Cifar-10/Cifar-100.

On shared columns, Table 2 can be considered a subset of Table 19. As discussed in subsection 5.2, Carso tops the comparison in terms of adversarial accuracy, while maintaining a clean accuracy comparable with that of some purification-based models.

Table 18, instead, investigates the same comparison across Carso and its *classifier* model, for adversarially-pretrained networks different from those described in section 5.1 – in particular, worse-performing. As it is possible to see, the increase in *end-to-end* adversarial accuracy determined by Carso does not depend in absolute terms on the quality of the original *classifier* model employed.

Table 18: Clean (results in *italic*) and adversarial (results in upright) accuracy for additional models to those described in subsection 5.2. The following abbreviations are used: `AT/Cl`: clean accuracy for the adversarially-pretrained model used as the *classifier*, when considered alone; `C/Cl`: clean accuracy for the Carso architecture; `AT/AA`: robust accuracy (by the means of AutoAttack) for the adversarially-pretrained model used as the *classifier*, when considered alone; `C/randAA`: robust accuracy for the Carso architecture, when attacked *end-to-end* by AutoAttack for randomised defences.

| Dataset | Classification model | AT/Cl | AT/AA | C/Cl | C/randAA |
|---|---|---|---|---|---|
| Cifar-10 | Rebuffi et al. (2021) | *0.8733* | 0.6075 | *0.8152* | 0.7070 |
| | Gowal et al. (2020) | *0.8948* | 0.6280 | *0.8361* | 0.7335 |
| Cifar-100 | Rebuffi et al. (2021) | *0.6241* | 0.3206 | *0.5723* | 0.5580 |

---

[4] At the time of paper review, Git SHA hash: `78fcc9e48a07a861268f295a777b975f25155964`.

Table 19: Clean (results in *italic*) and adversarial (results in upright) accuracy for *state-of-the-art* adversarial defences, compared to CARSO. The *Robust Acc.* column shows the best (*i.e.* lowest practically achieved) accuracy on test-set adversarial inputs, as obtained by either the original publication introducing the method, evaluation by AUTOATTACK as shown on the ROBUSTBENCH leaderboard, evaluation of bespoke adaptive attacks as shown on the ROBUSTBENCH leaderboard, or (for purification methods) evaluation by PGD+EOT from Lee & Kim (2024). The *Def. Type* column indicates whether the defence is based on adversarial training (`AT`), purification (`P`), or both (`AT+P`). Results for CARSO are the same as for Table 2.

| Dataset | Model | Architecture | Clean Acc. | Robust Acc. | Def. type |
|---|---|---|---|---|---|
| CIFAR-10 | Bartoldson et al. (2024) | WIDERESNET-94-16 | *0.9368* | 0.7371 | AT |
| | Amini et al. (2024) | MeanSparse WIDERESNET-94-16 | ***0.9568*** | 0.7310 | AT |
| | Peng et al. (2023) | RAWIDERESNET-70-16 | *0.9311* | 0.7107 | AT |
| | Wang et al. (2023) | WIDERESNET-70-16 | *0.9325* | 0.7069 | AT |
| | Bai et al. (2024b) | RESNET-152 + WIDERESNET-70-16 | *0.9519* | 0.6971 | AT |
| | Lin et al. (2024) | Diffusion-based | *0.9082* | 0.6641 | P |
| | Lee & Kim (2024) | Diffusion-based | *0.9053* | 0.5688 | P |
| | Nie et al. (2022) | Diffusion-based | *0.9043* | 0.5113 | P |
| | Hill et al. (2021) | Energy-based | *0.8412* | 0.5490 | P |
| | Yoon et al. (2021) | Diffusion-based | *0.8612* | 0.3711 | P |
| | Ours | CARSO (WIDERESNET-28-10) | *0.8686* | **0.7613** | AT+P |
| CIFAR-100 | Wang et al. (2023) | WIDERESNET-70-16 | *0.7522* | 0.4266 | AT |
| | Amini et al. (2024) | MEANSPARSE WIDERESNET-70-16 | *0.7513* | 0.4225 | AT |
| | Bai et al. (2024b) | RESNET-152 + WIDERESNET-70-16 | *0.8308* | 0.4180 | AT |
| | Cui et al. (2023) | WIDERESNET-28-10 | *0.7385* | 0.3918 | AT |
| | Bai et al. (2024a) | RESNET-152 + WIDERESNET-70-16 + MIXING NET | ***0.8521*** | 0.3872 | AT |
| | Lin et al. (2024) | Diffusion-based | *0.6973* | 0.4609 | P |
| | Ours | CARSO (WIDERESNET-28-10) | *0.6806* | **0.6665** | AT+P |

