# OpenReview forum: "Blending adversarial training and representation-conditional purification via aggregation improves adversarial robustness"
_TMLR — Accepted by TMLR_

### Review · Reviewer_jAY5 · 2025-03-28

**Summary Of Contributions:**

This paper proposes a new defense method against adversarial examples. Specifically, the paper suggests blending two popular approaches for adversarial defenses, i.e., *adversarial training* and *adversarial purification*. The method starts with training an adversarially-trained robust classifier, then a purifier is trained to generate a purified image from the adversarial example and the robust classifier’s internal representation (i.e., the activations from the adversarial input). The robust classifier makes the final classification on the purified image. With experiments, the paper demonstrates the improvement from using both adversarial training and adversarial purification.

**Audience:**

Yes

**Broader Impact Concerns:**

I don’t see a particular broader impact concern regarding this paper.

**Claims And Evidence:**

No

**Requested Changes:**

1. Add more experiments. Particularly, the paper should be clear about why both adversarial training and purification are essential for improvement.
2. Test the defense against more attack methods.
3. There are many existing adversarial training and purification-based defense methods. Choose a few representative baseline methods and compare the robust accuracies against them.
4. I don’t see enough justification about why adversarial training is needed for the proposed method. It would be worth experimenting with non-robust classifiers (in internal representation extraction, final classification, or both).

**Strengths And Weaknesses:**

### Strengths
1. To the best of my knowledge, the proposed method and architecture seem novel.
2. The paper contains many discussions justifying the design choices.
3. The experiment demonstrated remarkable improvements from the baseline.

### Weaknesses
1. This paper mainly proposes and evaluates a new defense method. I expect more various and extensive experimental studies for this type of paper, e.g., ablation study or hyperparameter search.
2. The paper reports robust accuracy against only one attack, AutoAttack. However, the proposed method should be challenged against other attacks, at least FGSM and PGD used in the training.
3. The paper claims that the method blends adversarial training and adversarial purification. However, the role of adversarial training seems unclear, and it might not be an essential component. If the purifier exploited the pattern in the activations, it should be able to use the activation patterns from a vanilla classifier.
4. It is unclear whether the purifier actually *purifies* the adversarial perturbation or it is just another sample that is convenient for the specific adversarially-trained classifier. To claim that it effectively removes adversarial perturbation, its accuracy on other classifiers could be tested.
5. The experiment should compare the improvement against the existing methods. To explain, the proposed method uses a specific variation of adversarial training, such as adversarially balanced batches. The resulting classifier could be significantly different from the robust classifier trained from existing adversarially-trained methods. While Table 2 shows the improvement of the accuracies, we cannot conclude that the proposed method achieves better robustness compared to existing defense methods.

---

> ### Author Response · Authors · 2025-06-27
> **Official Comment to Reviewer jAY5 (part 1 of 3)**
>
> We thank the Reviewer for their observations, which significantly contributed to improving and expanding the experimental scope, and clarify the exposition, of our work.
>
> #### About the *Weaknesses* section
>
> We will address the observations of the Reviewer outlined in the *Weaknesses* section in a similar list-based fashion.
>
> 1) **On the lack of further experiments**
>    As the Reviewer kindly suggested, we expanded the scope of our experiments to include a full ablation study on the WideResNet-28-10 models, trained on the CIFAR-10 and CIFAR-100 datasets.
>    In particular, we compared the following model setups.
>
>    **a)** An architecture identical to that proposed in the paper, where both image classifiers (*i.e.* the initial and the final one) are trained on *clean* examples only;
>
>    **b)** An architecture identical to that proposed in the paper, where the former image classifier (*i.e.* that used as a featuriser for the conditioning) is adversarially-trained, whereas the latter is not.
>
>    **c)** An architecture identical to that proposed in the paper, where the former image classifier is not adversarially-trained, but the latter is.
>
>    The case where a purifier is directly applied to the input image, and the purified reconstruction is classified by a non-adversarially-trained model is what we refer to as *purification* defence (in Table 2 of the original paper, and in the following). The case where an adversarially-trained model directly classifies the input is what we refer to as *adversarial training* defence (in Table 2 of the original paper, and in the following).
>
>     The results of such comparison are shown in the tables that follow, and will be integrated in the revised version of our manuscript.
>
>     As conventional within our original paper, the table heading *Clean Acc.* refers to test-set accuracy evaluated on non-corrupted datapoints, whereas the heading *Adv. Acc.* refers to accuracy computed on inputs perturbed by *randAA* (AutoAttack suitable for stochastic defences).
>
>     | CIFAR-10                  | Clean Acc. | Adv. Acc. |     |
>     | ------------------------- | ---------- | --------- | --- |
>     | Model a                   | 0.7314     | 0.7070    |     |
>     | Model b                   | 0.6743     | < 0.7070  |     |
>     | Model c                   | 0.8892     | 0.7975    |     |
>     | Best purification         | 0.9082     | 0.6641    |     |
>     | Best adversarial training | 0.9368     | 0.7371    |     |
>     | CARSO (as in the paper)   | 0.8686     | 0.7613    |     |
>
>     | CIFAR-100                 | Clean Acc. | Adv. Acc. |     |
>     | ------------------------- | ---------- | --------- | --- |
>     | Model a                   | 0.4395     | 0.4032    |     |
>     | Model b                   | 0.4373     | < 0.4032  |     |
>     | Model c                   | 0.6876     | 0.6716    |     |
>     | Best purification         | 0.6973     | 0.4609    |     |
>     | Best adversarial training | 0.7522     | 0.4267    |     |
>     | CARSO (as in the paper)   | 0.6806     | 0.6665    |     |
>
>     As it is possible to see, only the model in setup *(c)* does provide *clean* and *adversarial* accuracies comparable to that of CARSO - and indeed, it determines an improvement on both sides, for both datasets.
>
>     For that reason, we performed the evaluation in *Model c* setup on the TinyImageNet-200 dataset, which produced a comparable results (*w.r.t* CARSO) in terms of *clean* accuracy, but a 0.75% decrease in *robust accuracy*.
>
>     Keeping scalability to larger datasets (of which TinyImageNet-200 is the closest representative considered) in mind, and that any empirical robust accuracy result constitutes an upper bound of *maximally attainable* robust accuracy, we are convinced that the CARSO architecture proposed in the paper is overall the single most reliable and transparently auditable solution across the widest range of evaluation scenarios that produces a significant improvement in adversarial robust accuracy, at the cost of a slight *clean* accuracy penalty.
>
>     Since an adversarially-trained classifier would be used nonetheless for the final aggregated classification task, both in CARSO and *Model c* setup, the two do not differ in terms of training computational intensity. In case the reader is interested only in the CIFAR-10/100 scenarios, they will find the better-performing *Model c* variant described in the paper.

---

> > ### Author Response · Authors · 2025-06-27
> > **Official Comment to Reviewer jAY5 (part 2 of 3)**
> >
> > 2) **On the variety of test-time adversarial attacks**
> >    We understand the concern of the Reviewer *w.r.t.* testing CARSO against the widest array of adversarial attacks. As mentioned in *Subsection 5.1* of our original paper, although it is only *one* by name, AutoAttack is constituted by the worst-case scenario against an ensemble of adaptive adversarial attacks. In particular:
> >    - For the non-stochastic defence scenario: APGD against the *Cross-Entropy* loss, APGD against the *Difference of Logits Ratio* loss, FAB, and Square. Specifically, APGD is a generalisation and improvement upon PGD, which is in turn a generalisation and improvement upon FGSM.
> >    - For stochastic defences: APGD against the *Cross-Entropy* loss and APGD against the *Difference of Logits Ratio* loss, both whose gradients are averaged upon many runs of Expectation over Transformation (EoT). The FAB and Square adversarial attacks are unsuitable for their use with EoT, which in turn makes them unsuitable for attacking stochastic adversarial defences.
> >
> >     Additionally, as known from *Croce et al. (2020)* and *Cina' et al. (2025)*, testing an adversarial defence against attacks that are not optimally capable of producing the worst-case perturbation is not only scarcely informative, but also to blame of providing a false sense of security. We can safely assert, standing upon theory, that any assessment of CARSO conducted against PGD, FGSM, or APGD in a standardised fashion would not produce worse robustness performance by the assessed model as that measured by AutoAttack, according to the same standard. This has lead to the use of AutoAttack as a reference within the empirical adversarial robustness community. We follow such consensus.
> >
> >     However, since concerns have been specifically raised about the use of AutoAttack as compared to a specific variant of PGD+EoT in the context of diffusion/score-based purification methods *(Lee & Kim, 2024)*, we explicitly tested CARSO against that variant (called *PGD+EoT* in Table 2 of our original paper). Such test confirmed the robustness of CARSO, and the better effectiveness of AutoAttack in assessing it.
> >
> > 3) **On the role of adversarial training in CARSO**
> >    The Reviewer is correct in postulating that the representation-conditional purification machinery introduced with CARSO is able to provide an improved defence against adversarial attacks, even when activations are extracted from a cleanly-trained classifier. This is indeed what happens in the *Model c* setup mentioned above (*Answ. to Weaknesses*, 1) for the CIFAR-10/100 datasets. In such case, it is even able to produce better *clean* and *adversarial* accuracies *w.r.t.* the original CARSO. This is no longer the case for TinyImageNet-200, where representation extraction from a robust classifier results in improved overall robust accuracy. The final classifier, instead, must remain adversarially-trained to achieve significant robustness overall.
> >
> >    However, the blending of *purification* and *adversarial training* is not strictly limited to the use of *adversarially-trained* **classifiers**. In fact, whereas *purification-based* methods - as conventionally intended (see *Nie et al., 2022*, *Lin et al., 2024*) - learn from *clean* data a *clean* data manifold over which corrupted examples are reprojected for purification, our approach casts the problem directly during training as *denoising*. Since such process requires the generation of new adversarial examples against a part of the whole CARSO architecture (the initial classifier, kept frozen), this also contributes by itself to a blending of AT and purification.
> >
> > 4) **On the actual *purification ability* of the *purifier***
> >    We agree with the Reviewer. However, we would like to point out that we never claim within our paper that the *purifier* actually *purifies* adversarial examples, beyond its use as part of the CARSO architecture. And we do not want to. The name specifically refers to the input-to-input transformation of corrupted examples, before actual classification occurs - as popularised by *Nie et al., 2022*.
> >    To expand on the Reviewer's comment, it is not possible that the *purifier* adapts to the the specific classifier placed after it, as part of its training process - since such training does not involve at all the final classifier and is only driven by reconstruction loss. However, the choice of such classifier does influence the overall robustness of CARSO, as we can see from the ablation study shown above (*Answ. to Weaknesses*, 1).

---

> > > ### Author Response · Authors · 2025-06-27
> > > **Official Comment to Reviewer jAY5 (part 3 of 3)**
> > >
> > > 5) **On the comparison to existing methods**
> > >    We would like to point out that, wherever in our paper an *adversarially-trained* classifier is mentioned, or it is used as part of CARSO or otherwise, we refer to its standard version, as released to the public by the Authors of the corresponding reference publication (see *Subsection 5.1* of our original paper). In fact, we never performed the adversarial training of **classifiers** ourselves, rightly to avoid the (sometimes accidental) introduction of nonstandard variations *w.r.t.* the original method, which would have resulted in a moot comparison as the Reviewer highlights.
> > >    The *"specific variations of adversarial training, such as adversarially balanced batches"* the Reviewer mentions refer only to the representation-conditional purifier we introduce (see *Subsections 4.3, 4.4* of our paper), which has no reference or standardised equivalent from published literature. The justification of those variations, in such case, is provided in *Subsections 4.3, 4.4* and in *Appendix C* of our original paper.
> > >
> > > #### About the *Requested changes* section
> > >
> > > We will address the requests brought on by the Reviewer in the *Requested changes* section in a similar list-based fashion.
> > >
> > > 1) **Addition of more experiments**
> > >    See *Answ. to Weaknesses*, 1.
> > >
> > > 2) **Test against more attacks**
> > >    See *Answ. to Weaknesses*, 2
> > >
> > > 3) **Comparison against more defence methods**
> > >    *Table 2* of our original paper does compare the robust and clean accuracy of CARSO, against the best adversarial defences available at the time of paper submission - subdivided among *adversarial training*-based and *purification*-based. CARSO surpasses the best of both categories in terms of adversarial robustness, for all datasets/tasks considered. By the transitive property, it is guaranteed that CARSO does indeed perform equally well or better against worse-performing defences. Nonetheless, we will include a further table in an Appendix of the revised version of our manuscript, bearing *clean* and *adversarial* accuracy of top-performing defences from the RobustBench leaderboard (*adversarial training*-based) and from *Lee & Kim, 2024* (*purification*-based, properly evaluated). According to a comparison with our method, CARSO still tops the table in terms of adversarial accuracy, while maintaining a clean accuracy comparable with that of some purification-based models.
> > >
> > > 4) **Experiments with non-adversarially-trained classifiers**
> > >    See *Answ. to Weaknesses*, 1.
> > >
> > > ##### References
> > >
> > > - Croce et al., 2020: *RobustBench: a standardised adversarial robustness benchmark*;
> > > - Cina' et al, 2025: *Attackbench: Evaluating gradient-based attacks for adversarial examples*;
> > > - Lee & Kim, 2024: *Robust evaluation of diffusion-based adversarial purification*;
> > > - Nie et al., 2022: *Diffusion models for adversarial purification*;
> > > - Lin et al., 2024: *Robust diffusion models for adversarial purification*.

---

### Review · Reviewer_Hjaq · 2025-04-15

**Summary Of Contributions:**

The authors propose a adversarial defense strategy to strengthen the robustness of a fixed classifier $f$. At inference time, for a corrupted test input $x$, the predicted label is computed by aggregating predictions of the classifier $f$ on reconstructed samples $x_{1}, \ldots , x_{N}$. The reconstructed samples $x_i$ are generated using a stochastic conditional generative model (in this case conditional-VAEs) that utilize intermediate layer representations of the classifier $f$ on input $x$. To summarize, the authors propose a defense based on random reconstructions much like _purification_ strategies based on diffusion models [1].

The key insight of this article is that the intermediate neural representations of a pretrained classifier $f$ on a corrupted input $x$ provide valuable information to reconstruct clean inputs $\tilde{x}$ on the data manifold. This insight on its own is not novel (see [2]). Evidently if the classifier $f$ is poor a fit for the data or non-robust, then the representations at a perturbed input are unstable and less informative. Hence this procedure hinges on the quality of the classifier $f$. In this article, the authors use an off-the-shelf SOTA adversarially robust classifier $f$.

As per my understanding, the contribution of the article is -
(1) A specific architectural choice of a conditional-VAE for the stochastic generative model
(2) Experimental setups for training the conditional-VAE

This reviewer finds the framing "_blending adversarial training and adversarial purification strategies_" an overly complicated perspective of the work since the pretrained classifier is untouched - so no active interaction with adversarial training strategies.





References
1. Nie, W., Guo, B., Huang, Y., Xiao, C., Vahdat, A., & Anandkumar, A. Diffusion Models for Adversarial Purification.
2. Pouya Samangouei, Maya Kabkab, and Rama Chellappa. Defense-GAN: Protecting classifiers against adversarial
attacks using generative models.

**Audience:**

Yes

**Broader Impact Concerns:**

This paper does not raise any specific broader concern.

**Claims And Evidence:**

Yes

**Requested Changes:**

1. __Experimental results on Imagenet.__
Such a result would strengthen the paper since scalability is a critical concern for adversarial purification strategies. If computational resources are a concern, the authors could demonstrate results on subsets of Imagenet (for eg. with 100 classes).  However I don't think this alone is sufficient to reject this article.

2. __Experiments on diverse pretrained classifiers.__
I believe the experiments are run for a fixed pretrained WideResNet-28-10 from Cui et al. (2023) and  Wang et al. (2023). The authors can bolster their message by running it on multiple SOTA robust classifiers from the RobustBench leaderboard.

**Strengths And Weaknesses:**

## Strengths
- The proposed appears to improve the robustness of the pretrained robust classifiers.


## Weakness
The article is sparse on hard theoretical facts or justifications. At present, it reads as a specific combination of ideas that already exist in literature.
- If this article is accepted its because the experiments show improved adversarial robustness due to the proposed defense. However, there is no mechanistic interpretation or understanding of why there is an improvement, in particular, it is unclear for what learning tasks the proposed method can be expected to work.
- The authors employ an unorthodox choice of robust aggregation using double exponentials. While the appendix has a heuristic justification, I would like to understand a simple setting where the proposed aggregation mechanism is the clear correct choice. Further, how does this choice compare to an alternate naive choice - predicting the majority label over the reconstructed inputs $x_i$ ?


#### Minor Issues
- The authors frequently employ \emdash and hyphens to interleave context in the middle of sentences. This reviewer found some sentences hard to parse.
- The discussion of adversarial training/PGD attack schemes are unnecessary (given they are exact instantiations of established procedures)

---

> ### Author Response · Authors · 2025-06-27
> **Official Comment to Reviewer Hjaq (part 1 of 5)**
>
> We thank the Reviewer for their observations, which significantly contributed to improving and expanding the experimental scope, and clarify the exposition, of our work.
>
> #### About the *Summary Of Contributions* section
>
> We would like to thank the Reviewer for having taken time to analyse our contribution, whose summary is precise and captures the spirit of our paper. However, we would argue that, architectural choices aside, significant differences are present between CARSO and mainstream diffusion/score-based *purification* defences such as those from *Nie et al., 2022* or *Lin et al., 2024*. Such differences provide to CARSO both novelty and a more reliable robustness evaluation. Specifically:
> - Typical *purification-based* defences are trained on *clean* input generative modelling, using an unperturbed example dataset: at inference time, noise is diffused in corrupted inputs, and the result reconstructed to a *clean* example. On the other hand, CARSO directly models adversarial input purification as a perturbed-to-clean (as well as clean-to-clean) denoising process, during training.
> - Typical *purification-based* defences are trained on input modelling directly on input data. Our strategy integrates - for the first time to the best of our knowledge - a conditioning step on the internal representation of a frozen classifier. Such internal representation is, at inference time, the only link between the original (potentially corrupted) input and the reconstructed example.
> - Typical *purification-based* defences are meant to be directly followed by a standard classification model, learned on *clean* examples only. This is usually required, since the use of an adversarially-trained classifier would become the limiting factor in the overall adversarial accuracy attainable by such setup. This is not the case of CARSO, as the ablation study provided *e.g.* in answers to Reviewers `jAY5` and `xTRE` (which will be included in the revised version of the manuscript, and shown at the end for this answer completeness) shows that the use of a *final* adversarially-trained classifier is crucial to achieve the robust accuracy shown by CARSO.
> - Finally, CARSO, unlike diffusion/score-based *purification* defences, is not vulnerable to robustness overconfidence when assessed by means of AutoAttack (a phenomenon originally discovered by *Lee & Kim, 2024*), as shown in *Table 2* and *Section 5.2, (a)* of our original paper.
>
> We agree with the remark that *"\[our\] article \[shows\] that the intermediate neural representations of a pretrained classifier on a corrupted input provide valuable information to reconstruct clean inputs on the data manifold"*. However, we defend that the implementation of such insight proposed with CARSO is novel to the best of our knowledge. We have also been unable to find references to neither this insight, nor to such an implementation in the paper referred to as prior art (*Samangouei et al., 2018*).
>
> The Reviewer also mentions that the overall robustness of CARSO may hinge  on the quality of the initial classifier - and that if such classifier is non-robust to input corruptions, this may result in subpar performance. However, the phenomenon is not so clear-cut.  The aforementioned ablation study shows that this is not universally true: in particular, in the case of small datasets / non overly complex tasks (CIFAR-10/100) the use of a non-robust classifier for representation extraction may result in improved *clean* as well as *robust* performance. This becomes no longer true for more complex datasets/tasks, such as TinyImageNet-200, where the initial classifier must be adversarially-trained in order to achieve the most robust model.
>
> As mentioned in *Subsection 4.1* of our original paper, most of the architectural choices operated in the design of CARSO are not intrinsically tied to the use of a conditional VAE for purification: any model capable of stochastic conditional data generation at inference time would suffice. As discussed in the same subsection, the choice of a (conditional) VAE has been determined by the need to assess CARSO in the most favourable scenario for an attacker, and the least favourable for the defender. In fact, any empirical adversarial assessment can only provide an upper bound to *maximally attainable* robust accuracy. The experimental assessment follows such principle.

---

> > ### Author Response · Authors · 2025-06-27
> > **Official Comment to Reviewer Hjaq (part 2 of 5)**
> >
> > Finally, as far as the framing of our method as a *blending of AT and purification*, we would like to point out that such blending is not strictly limited to the use of *adversarially-trained* **classifiers**. In fact, whereas *purification-based* methods - as conventionally intended (see *Nie et al., 2022*, *Lin et al., 2024*, and above) - learn from *clean* data a *clean* data manifold over which corrupted examples are reprojected for purification, our approach casts the problem directly during training as *denoising*. Since such process requires the generation of new adversarial examples against a part of the whole CARSO architecture (the initial classifier, kept frozen), this also contributes by itself to a blending of AT and purification.
> >
> > #### About the *Weaknesses* section
> >
> > We will address the observations of the Reviewer outlined in the *Weaknesses* section, where suitable, in a similar list-based fashion.
> >
> > As the article is structured, whenever a design choice has been made during the development of CARSO, the thought process leading to such choice and - if available - a more formal justification are provided.
> > This is the case of *Subsection 4.1* of our original paper (where the choice of a conditional VAE as a purification model is explained), *Subsection 4.2* (where the general guiding principles for the architectural design of CARSO are explained), *Subsections 4.3, 4.4, 4.5* and *Appendix D* (where justification to each of the non-standard architectural and training choices operated throughout the paper is provided).
> > Most importantly, as commonplace for empirical adversarial robustness works, we consider and propose standardised assessment against strong adaptive adversarial attacks with an $\ell_\infty$ perturbation bound as the ultimate corroboration that our method can be used effectively. As such, we agree that our contribution is mostly experimental, although guided by some first-principles considerations.
> >
> > As to why we do not consider CARSO (and, by extension, our paper) a simple *mix-and-match* of existing methods published in literature, we refer to our previous, lengthy analysis of the *Summary Of Contribution* section.
> >
> > 1) We agree with the Reviewer that our work does not provide a direct causal or mechanistic explanation for the improved defensive capability of CARSO. Our work has been thought as fundamentally experimental in spirit, with a large portion of overall efforts and space dedicated to transparency and reliability in experimental evaluation. This intention, *e.g.*, explains the choice of assessing the method in a deliberately difficult scenario for the defender (the choice of the VAE, the use of worst-case attacks, the additional test against PGD+EoT), the sanity check against gradient obfuscation, and a more analytical comparison of aggregation strategies. Additionally, the ablation study suggested by other Reviewers (which will be included in the revised manuscript) further contributes in such direction. For a justification (although nor analytical) of our design choices, we refer to *Subsection 4.1* of our original paper (where the choice of a conditional VAE as a purification model is explained), *Subsection 4.2* (where the general guiding principles for the architectural design of CARSO are explained), *Subsections 4.3, 4.4, 4.5* and *Appendix D* (where justification to each of the non-standard architectural and training choices operated throughout the paper is provided).
> >
> >    In any case, there is no intrinsic limitation to the applicability of CARSO to tasks whose inputs can be modelled generatively, and whose target can be cast as a classification problem. In our experimental assessment, as commonplace, we framed the testing as pertaining to image classification problems; we disclose such framing since the abstract of our paper.

---

> > > ### Author Response · Authors · 2025-06-27
> > > **Official Comment to Reviewer Hjaq (part 3 of 5)**
> > >
> > > 2) Unfortunately, we did not develop an analytical theory in justification of the specific aggregation function used as part of CARSO - although we agree that it would much strengthen its theoretical grounding beyond heuristics, and promote its cross-utilisation in different scenarios.
> > >
> > >    With respect to the most intuitive *majority-vote* aggregation, which was the first we considered in the early phases of our work, the most relevant limiting factor is the lack of exact differentiability. In fact, by proposing the use of majority-vote aggregation, we would have inevitably had to choose among either being unable to properly assess the vulnerability of the system (any *white-box* result would have been the effect of gradient obfuscation) or being required to perform such assessment against an approximated model (using *e.g.* straight-through gradient estimation, or a different, differentiable aggregation function).
> > >
> > >    In any case, we show in the tables below the *clean* and *robust* accuracy results for the AutoAttack assessment of the model with majority-vote aggregation. The models and hyperparameters are exactly the same as those used in the paper.
> > >
> > >     As conventional within our original paper, the table heading *Clean Acc.* refers to test-set accuracy evaluated on non-corrupted datapoints, whereas the heading *Adv. Acc.* refers to accuracy computed on inputs perturbed by *randAA* (AutoAttack suitable for stochastic defences). The last column of the table (*Unreliable Gradient Samples*) indicates the fraction of overall gradient evaluations by AutoAttack that resulted in an exactly zero or `NaN` gradient, and that the implementation employs as a builtin sanity check. Any number of samples significantly larger than zero indicates unreliability of the method.
> > >
> > >     | CIFAR-10                  | Clean Acc. | Adv. Acc. | Unr. Grad. Samples |    |
> > >     | ------------------------- | ---------- | --------- | ------------------ | -- |
> > >     | CARSO                     | 0.8686     | 0.7613    | 0%                 |    |
> > >     | Maj. Vote Aggregation     | 0.8691     | 0.8602    | > 99%               |    |
> > >
> > >     | CIFAR-100                 | Clean Acc. | Adv. Acc. | Unr. Grad. Samples |    |
> > >     | ------------------------- | ---------- | --------- | ------------------ | -- |
> > >     | CARSO                     | 0.6806     | 0.6665    | 0%                 |    |
> > >     | Maj. Vote Aggregation     | 0.6805     | 0.6698    | > 99%               |    |
> > >
> > >     As it is possible to see by comparing the *clean* accuracies from the table above, and those from *Appendix D*, we can establish that the unorthodox doubly-exponential aggregation does not deviate from more intuitive *logit aggregation* or *probability aggregation* more than majority-vote aggregation.
> > >
> > > ###### Minor issues
> > >
> > > 1) **On the use of hyphenated parenthetical clauses**
> > >   We will revise some stylistic choices for the manuscript in order to improve overall readability. We thank the Reviewer for such remark.
> > >
> > > 2) **On the description of PGD adversarial training**
> > >   Within the main text of the paper, only 8 lines are dedicated to the discussion of PGD AT. Of those, 1 introduces the reference work on the topic, while 2 introduce some notation used throughout the paper. Most of the description of the method is relegated to *Appendix A*, which could be potentially removed. The same can also be said of VAEs and conditional VAEs, which are briefly introduced in *Section 3* and described more lengthily in *Appendix B*.
> > >
> > > ###### Ablation Study
> > > We show here the results of the ablation study mentioned above, on the WideResNet-28-10 models trained on the CIFAR-10 and CIFAR-100 datasets, which will be integrated in a revised version or our manuscript.
> > >
> > > In particular, we compared the following model setups.
> > >
> > > **a)** An architecture identical to that proposed in the paper, where both image classifiers (*i.e.* the initial and the final one) are trained on *clean* examples only;
> > >
> > > **b)** An architecture identical to that proposed in the paper, where the former image classifier (*i.e.* that used as a featuriser for the conditioning) is adversarially-trained, whereas the latter is not.
> > >
> > >  **c)** An architecture identical to that proposed in the paper, where the former image classifier is not adversarially-trained, but the latter is.
> > >
> > >  The case where a purifier is directly applied to the input image, and the purified reconstruction is classified by a non-adversarially-trained model is what we refer to as *purification* defence (in Table 2 of the original paper, and in the following). The case where an adversarially-trained model directly classifies the input is what we refer to as *adversarial training* defence (in Table 2 of the original paper, and in the following).
> > >
> > > The results of such comparison are shown in the tables that follow, and will be integrated in the revised version of our manuscript.

---

> > > > ### Author Response · Authors · 2025-06-27
> > > > **Official Comment to Reviewer Hjaq (part 4 of 5)**
> > > >
> > > > As conventional within our original paper, the table heading *Clean Acc.* refers to test-set accuracy evaluated on non-corrupted datapoints, whereas the heading *Adv. Acc.* refers to accuracy computed on inputs perturbed by *randAA* (AutoAttack suitable for stochastic defences).
> > > >
> > > > | CIFAR-10                  | Clean Acc. | Adv. Acc. |
> > > > | ------------------------- | ---------- | --------- |
> > > > | Model a                   | 0.7314     | 0.7070    |
> > > > | Model b                   | 0.6743     | < 0.7070  |
> > > > | Model c                   | 0.8892     | 0.7975    |
> > > > | Best purification         | 0.9082     | 0.6641    |
> > > > | Best adversarial training | 0.9368     | 0.7371    |
> > > > | CARSO (as in the paper)   | 0.8686     | 0.7613    |
> > > >
> > > > | CIFAR-100                 | Clean Acc. | Adv. Acc. |
> > > > | ------------------------- | ---------- | --------- |
> > > > | Model a                   | 0.4395     | 0.4032    |
> > > > | Model b                   | 0.4373     | < 0.4032  |
> > > > | Model c                   | 0.6876     | 0.6716    |
> > > > | Best purification         | 0.6973     | 0.4609    |
> > > > | Best adversarial training | 0.7522     | 0.4267    |
> > > > | CARSO (as in the paper)   | 0.6806     | 0.6665    |
> > > >
> > > > As it is possible to see, only the model in setup *(c)* does provide *clean* and *adversarial* accuracies comparable to that of CARSO - and indeed, it determines an improvement on both sides, for both datasets.
> > > >
> > > >  For that reason, we performed the evaluation in *Model c* setup on the TinyImageNet-200 dataset, which produced a comparable results (*w.r.t* CARSO) in terms of *clean* accuracy, but a 0.75% decrease in *robust accuracy*.
> > > >
> > > > Keeping scalability to larger datasets (of which TinyImageNet-200 is the closest representative considered) in mind, and that **any** empirical robust accuracy result constitutes an upper bound of *maximally attainable* robust accuracy, we are convinced that the CARSO architecture proposed in the paper is overall the single most reliable and transparently auditable solution across the widest range of evaluation scenarios that produces a significant improvement in adversarial robust accuracy, at the cost of a slight *clean* accuracy penalty.
> > > >
> > > > Since an adversarially-trained classifier would be used nonetheless for the final aggregated classification task, both in CARSO and *Model c* setup, the two do not differ in terms of training computational intensity. In case the reader is interested only in the CIFAR-10/100 scenarios, they will find the better-performing *Model c* variant described in the paper.
> > > >
> > > > #### About the *Requested changes* section
> > > >
> > > > 1) We agree with the Reviewer that properly-conducted experiments on ImageNet would strengthen the paper and its claims. However, due to resource constraints, we have been unable to perform them rigorously. In fact, not only such scale-up would have determined an increase in classification model size (twice, *i.e.* for both the initial and final classifier) in order to attain classification accuracies worth of notice. Also, it would have required a massive scale-up of the representation-extraction machinery and the purified image generator. Additionally, in the specific form of a conditional VAE, it could not have had sufficient expressive power to model full-size ImageNet-like inputs. Lastly, proper testing of the resulting model by means of AutoAttack for stochastic defences would have taken significant time, even when executed in an accelerated and distributed fashion: in the experiments performed within the paper, and as a consequence of Reviewer's requests, testing time has taken the largest fraction of compute time, well adove 3/4ths.
> > > >
> > > >    The closest approximation among the datasets used in the paper is TinyImageNet-200. In such case, both input size ($64 \times 64 \times 3$ instead of $256 \times 256 \times 3$) and the number of classes ($200$ instead of $1000$) are reduced in comparison to the original. The images and class labels are obtained directly as a subset of the full ImageNet.
> > > > 2) We thank the Reviewer for their suggestion. In our paper, we test our approach using three pretrained classifier models, which are architecturally similar (and all three based upon a *Wide ResNet*-28-10), *i.e.:* the WRN-28-10 from *Cui et al., 2023* trained on CIFAR-10, that from *Cui et al., 2023* trained on CIFAR-100, and a slightly different WRN-28-10 from *Wang et al., 2023*, trained on TinyImageNet-200.

---

> > > > > ### Author Response · Authors · 2025-06-27
> > > > > **Official Comment to Reviewer Hjaq (part 5 of 5)**
> > > > >
> > > > > To provide additional scope to the experimental assessment, we selected three additional models from the RobustBench leaderboard, also in this case WRN-28-10s to avoid the re-engineering efforts required by the selection of internal representation layers while maintaining scale under control. It particular, we picked models from *Gowal et al., 2020* and *Rebuffi et al., 2021* whose pretrained weights are freely distributed by the Authors. Results are shown in the tables below.
> > > > >
> > > > > As conventional within our original paper, the table heading *AT/CL* (*C/CL*) refers to test-set accuracy of the adversarially-trained classifier (entire CARSO architecture) evaluated on non-corrupted datapoints, whereas the heading *AT/AA* (*C/randAA*) refers to accuracy computed on inputs perturbed, respectively by standard AutoAttack and *randAA* (AutoAttack suitable for stochastic defences).
> > > > >
> > > > >
> > > > >   | CIFAR-10                           | AT/CL  | AT/AA  | C/CL   | C/randAA |    |
> > > > >   | ---------------------------------- | ------ | ------ | ------ | -------- | -- |
> > > > >   | *Rebuffi et al., 2021* (WRN-28-10) | 0.8733 | 0.6075 | 0.8152 | 0.7070   |    |
> > > > >   | *Gowal et al., 2020* (WRN-28-10)   | 0.8948 | 0.6280 | 0.8361 | 0.7335   |    |
> > > > >
> > > > >   | CIFAR-100                          | AT/CL  | AT/AA  | C/CL   | C/randAA |     |
> > > > >   | ---------------------------------- | ------ | ------ | ------ | -------- | --- |
> > > > >   | *Rebuffi et al., 2021* (WRN-28-10) | 0.6241 | 0.3206 | 0.5723 | 0.558    |     |
> > > > >
> > > > >   These results corroborate once more the conclusions expressed in the paper.
> > > > >
> > > > >
> > > > > ##### References
> > > > > - Nie et al., 2022: *Diffusion models for adversarial purification*;
> > > > > - Lin et al., 2024: *Robust diffusion models for adversarial purification*;
> > > > > - Lee & Kim, 2024: *Robust evaluation of diffusion-based adversarial purification*;
> > > > > - Samangouei et al., 2018: *Defence-GAN: Protecting classifiers against adversarial attacks using generative models*;
> > > > > - Cui et al., 2023: *Decoupled Kullback-Leibler divergence loss*
> > > > > - Wang et al., 2023: *Better diffusion models further improve adversarial training*
> > > > > - Gowal et al., 2020: *Uncovering the Limits of Adversarial Training against Norm-Bounded Adversarial Examples*;
> > > > > - Rebuffi et al., 2021: *Fixing Data Augmentation to Improve Adversarial Robustness*.

---

### Review · Reviewer_xTRE · 2025-06-23

**Summary Of Contributions:**

This study proposes a framework called CARSO that effectively integrates adversarial training, adversarial purification, and randomization, which are existing powerful defense techniques against adversarial attacks. CARSO trains a Variational Auto Encoder (VAE)-based purifier while freezing an adversarially trained image classifier. During inference, it aggregates the predictions of multiple purified images to introduce randomness.
CARSO is evaluated through numerical experiments using three major datasets, including CIFAR10, CIFAR100, and Tiny ImageNet. Robustness evaluation is conducted using AutoAttack for randomized defenses and PGD+EOT because CARSO incorporates randomness. The proposed framework demonstrates higher robust accuracy compared to the robust accuracy of AT models after AutoAttack.

**Audience:**

Yes

**Claims And Evidence:**

No

**Requested Changes:**

Please see weaknesses above.

**Strengths And Weaknesses:**

Strengths
---

1. This framework, which significantly improves robust accuracy simply by adding a VAE-based purifier that can be trained quickly compared to the diffusion-based purifier, is practical and easy to incorporate with existing classifiers.

2. The evaluation employs attack methods that assume non-deterministic defenses, and the results have a certain level of reliability.

3. The authors describe the limitations of the proposed framework, providing readers with useful information to understand the applicability of CARSO.


Weaknesses
---

1. It is understandable that using a VAE trained with model features as conditions to purify images results in greater robustness, but I do not understand the necessity of using features from an adversarial training model. Please provide experimental or theoretical evidence to explain why it is necessary to train a VAE based on features from an adversarial training model.

2. The proposed method consists of three important components: adversarial training, purification, and aggregation. However, experiments to evaluate the contribution of each component are not sufficient. While evaluation experiments on the aggregation method are conducted in Appendix D, settings without adversarial training or with the purifier simply connected in series have not been evaluated, making it difficult to understand the role of each component.

---

> ### Author Response · Authors · 2025-06-27
> **Official Comment to Reviewer xTRE (part 1 of 2)**
>
> We thank the Reviewer for their observations, which contributed to improving and expanding the experimental scope of our work.
>
> #### About the *Weaknesses* section
>
> We will address the observations of the Reviewer outlined in the *Weaknesses* section in a similar list-based fashion. Since the concerns raised by the Reviewer mostly overlap with some remarks made by Reviewer `jAY5` (*Answ. to Weaknesses*, 1, 3), our answer will extensively borrow from that provided to Reviewer `jAY5`.
>
> 1) **On the necessity of an adversarially-trained classifier for representation extraction**
>    The Reviewer is correct in postulating that the representation-conditional purification machinery introduced with CARSO is able to provide an improved defence against adversarial attacks, even when activations are extracted from a cleanly-trained classifier.
>    As to why an adversarially-trained classifier is then used as part of our proposal, we conducted an ablation study on the WideResNet-28-10 models, trained on the CIFAR-10 and CIFAR-100 datasets.
>    In particular, we compared the following model setups.
>
>    **a)** An architecture identical to that proposed in the paper, where both image classifiers (*i.e.* the initial and the final one) are trained on *clean* examples only;
>
>    **b)** An architecture identical to that proposed in the paper, where the former image classifier (*i.e.* that used as a featuriser for the conditioning) is adversarially-trained, whereas the latter is not.
>
>    **c)** An architecture identical to that proposed in the paper, where the former image classifier is not adversarially-trained, but the latter is.
>
>    The case where a purifier is directly applied to the input image, and the purified reconstruction is classified by a non-adversarially-trained model is what we refer to as *purification* defence (in Table 2 of the original paper, and in the following). The case where an adversarially-trained model directly classifies the input is what we refer to as *adversarial training* defence (in Table 2 of the original paper, and in the following).
>
>     The results of such comparison are shown in the tables that follow, and will be integrated in the revised version of our manuscript.
>
>     As conventional within our original paper, the table heading *Clean Acc.* refers to test-set accuracy evaluated on non-corrupted datapoints, whereas the heading *Adv. Acc.* refers to accuracy computed on inputs perturbed by *randAA* (AutoAttack suitable for stochastic defences).
>
>     | CIFAR-10                  | Clean Acc. | Adv. Acc. |     |
>     | ------------------------- | ---------- | --------- | --- |
>     | Model a                   | 0.7314     | 0.7070    |     |
>     | Model b                   | 0.6743     | < 0.7070  |     |
>     | Model c                   | 0.8892     | 0.7975    |     |
>     | Best purification         | 0.9082     | 0.6641    |     |
>     | Best adversarial training | 0.9368     | 0.7371    |     |
>     | CARSO (as in the paper)   | 0.8686     | 0.7613    |     |
>
>     | CIFAR-100                 | Clean Acc. | Adv. Acc. |     |
>     | ------------------------- | ---------- | --------- | --- |
>     | Model a                   | 0.4395     | 0.4032    |     |
>     | Model b                   | 0.4373     | < 0.4032  |     |
>     | Model c                   | 0.6876     | 0.6716    |     |
>     | Best purification         | 0.6973     | 0.4609    |     |
>     | Best adversarial training | 0.7522     | 0.4267    |     |
>     | CARSO (as in the paper)   | 0.6806     | 0.6665    |     |
>
>     As it is possible to see, only the model in setup *(c)* does provide *clean* and *adversarial* accuracies comparable to that of CARSO - and indeed, it determines an improvement on both sides, for both datasets.
>
>     For that reason, we performed the evaluation in *Model c* setup on the TinyImageNet-200 dataset, which produced a comparable results (*w.r.t* CARSO) in terms of *clean* accuracy, but a 0.75% decrease in *robust accuracy*.
>
>     Keeping scalability to larger datasets (of which TinyImageNet-200 is the closest representative considered) in mind, and that any empirical robust accuracy result constitutes an upper bound of *maximally attainable* robust accuracy, we are convinced that the CARSO architecture proposed in the paper is overall the single most reliable and transparently auditable solution across the widest range of evaluation scenarios that produces a significant improvement in adversarial robust accuracy, at the cost of a slight *clean* accuracy penalty.
>
>     Since an adversarially-trained classifier would be used nonetheless for the final aggregated classification task, both in CARSO and *Model c* setup, the two do not differ in terms of training computational intensity. In case the reader is interested only in the CIFAR-10/100 scenarios, they will find the better-performing *Model c* variant described in the paper.

---

> > ### Author Response · Authors · 2025-06-27
> > **Official Comment to Reviewer xTRE (part 2 of 2)**
> >
> > 2) **On the individual role of *adversarial training*, *purification*, and *aggregation* within CARSO**
> >    The ablation study provided above and its results (*Answ. to Weaknesses*, 1), do provide an answer to the Reviewer's legitimate doubts. In particular, to recap and integrate the content of the previous answer:
> >    - The use of two *clean* classifiers, as initial feature extractor and final classifier, does result in worse defensive ability as opposed to CARSO.
> >    - The substitution of only the former classifier with an adversarially-trained one does not determine an increase, neither in terms of *clean*, nor of *robust* accuracy. As such, it cannot be pointed to, alone, as the reason for robustness increase in CARSO.
> >    - The substitution of only the latter classifier with an adversarially-trained one does instead improve upon the *clean*/*clean* setup. This remains advantageous even against CARSO, as long as the dataset/task remains of moderate size and complexity.
> >    - As dataset/task choice becomes more complex, the use of both *adversarially-trained* classifiers is necessary to obtain the greatest defensive capability.
> >    - As far as *adversarial training* alone is considered, we refer to the comparison of CARSO (and related models described in the previous answer) against the very same models used within CARSO, when attacked alone. Such comparisons are contained in *Table 2* of our paper (and in tables above).
> >    - As far as *adversarial purification* alone is considered, we did not perform tests where a VAE is simply juxtaposed before a clean/adversarially-trained classifier as *Gu & Rigazio, 2015* warn against such setup possibly leading to decreased robustness in comparison to an adversarial classifier alone. As such, it is possible to upper-bound by a significant margin the robust accuracy of this hypothetical setup with that of the sole adversarial classifier. Results from *Table 5* of *Lee & Kim, 2024* - which use *state-of-the-art* adversarially-trained models after *state-of-the-art* diffusion/score-based purification models confirm such intuition in all cases considered.
> >
> > #### About the *Requested changes* section
> >
> > See *Answ. to Weaknesses*.
> >
> > ##### References
> >
> > - Gu & Rigazio, 2015: *Towards deep neural network architectures robust to adversarial examples*;
> > - Lee & Kim, 2024: *Robust evaluation of diffusion-based adversarial purification*.

---

### Author Response · Authors · 2025-07-01
**Official Comment by Authors and Summary of Changes**

We thank all Reviewers for their comments and suggestions, which collectively contributed to an improvement in the overall rigour and clarity of our work.

We have just uploaded a revised version of our manuscript, containing all the *requested changes* we have been able to fulfil, as well as some additional minor edits and adjustments. The most relevant of those are summarised below.

- The title of the paper has been slightly changed, to make content and contribution clearer and more directly understandable. The new title is: `Blending adversarial training and representation-conditional purification via aggregation improves adversarial robustness`.
- Description and results of the ablation study about the need for adversarially-trained *classifier* and/or *purifier* (suggested by Reviewers `jAY5` and `xTRE`, and relevant to answer to all three Reviewers) have been added to (new) *Appendix D* of the revised version of the paper, and commented.
- A wider-spectrum comparison of CARSO against numerous top-performing existing defences from literature (suggested by Reviewer `jAY5`) has been added to (new) *Appendix E* of the revised version of the paper, and commented. Since the resulting tabular comparison is a superset of (was) *Appendix F* of the original paper (*Table 16*), the two appendices (and contained tables) have been merged and referred to accordingly.
- An additional table containing results from applying CARSO to additional pre-trained *classifiers* (as suggested by Reviewer `Hjaq`) has been included in (new) *Appendix E* of the revised version of the paper, and commented.
- An additional table showing the results from using *majority-vote* aggregation within CARSO (as suggested by Reviewer `Hjaq`) has been included in *Appendix B* of the revised version of the paper, and commented.
- Further references to literature have been added within some passages subject to major discussion with Reviewers, as to match relevant literature mentioned in the discussion.
- *Appendix A* and *Appendix B* of the original paper have been removed for the sake of brevity (as suggested by Reviewer `Hjaq`). They contained a summary of methods from existing literature (not novel, though used as part of our contribution). As a result, the letter-based enumeration of *Appendices* shifted. Particular care in drafting answers to Reviewers and this summary has been taken to always specify whether the appendix referred to follows the old or the new enumeration.

---

### Decision · Action_Editor_b9UH · 2025-08-23

**Recommendation:** Accept as is

**Audience:**

Yes

**Audience Explanation:**

Researchers who work in the area of adversarial robustness will find this paper useful.

**Claims And Evidence:**

Yes

**Claims Explanation:**

The paper introduces a novel defense method against adversarial examples, with promising experimental results across multiple setups. Reviewer 1 finds the observations convincing and recommends acceptance, noting that the lack of large-scale experiments such as ImageNet is understandable given computational constraints. Reviewer 3 also recognizes the potential of the proposed CARSO method, highlighting its effectiveness but requesting further validation on larger datasets to strengthen claims of generality. While Reviewer 2 expresses concerns about motivation and justification, the authors have clarified distinctions from prior purification strategies and addressed many points in their rebuttal. On balance, the novelty of the approach, soundness of experiments, and overall promise of the method outweigh the limitations. I recommend acceptance, while encouraging the authors to extend their evaluation on larger-scale datasets in future work.